# Trading off Image Quality for Robustness is not Necessary with Regularized Deterministic Autoencoders

**Amrutha Saseendran**[1], **Kathrin Skubch**[1], **Stefan Falkner**[1], and **Margret Keuper**[2,3]
[1]Bosch Center for Artificial Intelligence
[2]University of Siegen, [3]Max Planck Institute for Informatics, Saarland Informatics Campus
`Amrutha.Saseendran@de.bosch.com`

## Abstract

The susceptibility of Variational Autoencoders (VAEs) to adversarial attacks indicates the necessity to evaluate the robustness of the learned representations along with the generation performance. The vulnerability of VAEs has been attributed to the limitations associated with their variational formulation. Deterministic autoencoders could overcome the practical limitations associated with VAEs and offer a promising alternative for image generation applications. In this work, we propose an adversarially robust deterministic autoencoder with superior performance in terms of both generation and robustness of the learned representations. We introduce a regularization scheme to incorporate adversarially perturbed data points to the training pipeline without increasing the computational complexity or compromising the generation fidelity when compared to the robust VAEs by leveraging a loss based on the two-point Kolmogorov–Smirnov test between representations. We conduct extensive experimental studies on popular image benchmark datasets to quantify the robustness of the proposed approach based on the adversarial attacks targeted at VAEs. Our empirical findings show that the proposed method achieves significant performance in both robustness and fidelity when compared to the robust VAE models. An implementation is available at `https://github.com/boschresearch/Robust_GMM_DAE`.

## 1   Introduction

Variational autoencoders (VAEs) offer a powerful probabilistic framework to learn deep generative models. They are widely employed in various domains such as computer vision, natural language processing and representation learning [24, 16, 25, 26, 10]. A VAE is composed of two coupled but independently parameterized models, an encoder or a recognition model that computes a latent representation of the input space with stochastic latent variables and a decoder or a generative model to map the latent representations back to the original input space [18]. One of the major advantages of VAEs is that they provide semantically meaningful latent representations of high-dimensional complex input distributions which can be further utilized for various downstream tasks [17, 14, 28, 11]. However, VAEs are often limited due to theoretical and practical limitations such as over-regularization, prior-posterior mismatch resulting in trade-offs between generation and reconstruction fidelity [31, 9, 12]. Recently proposed deterministic versions of VAEs offer promising alternatives to overcome these limitations [12, 27].

The semantically meaningful representations learned by VAEs can still be corrupted by so-called adversarial attacks [30, 19, 20], where even small but specifically crafted changes to the input can lead to very different reconstructions. This observation reveals a lack of generalization within such models and is therefore a serious concern with respect to many practical applications. While it

36th Conference on Neural Information Processing Systems (NeurIPS 2022).

is harder to attack VAEs when compared to classifier networks [13] it is essential to analyze the robustness of VAEs along with their generative performance, to validate whether the learned latent representations are meaningful. Hence, there has been increasing research interest in the deep learning field towards training robust models, both for classifier models and for autoencoders, i.e. for robust representation learning. Borrowed from the training of robust classification models [22, 33], the concept of adversarial training has proven to be able to smoothen the VAE encoder and improve the robustness of the learned representations [6]. Other attempts toward learning robust representation spaces introduce either complex network architectures or expensive regularization mechanisms to improve the robustness of VAEs [32, 2]. Further, it has been pointed out in previous works that the robustness of VAEs can be improved by generating disentangled latent representations or by encouraging the smoothness or consistency of the encoding-decoding process [32, 5]. However, regularizing the VAE objective to enhance robustness often leads to poor generation ability compared to its non-robust counterpart. Hence, we seek to focus on improving the robustness of autoencoders while still maintaining the generation performance.

In this work, we introduce a simple and easy to train deterministic autoencoder which exhibits superior performance in terms of both generation and adversarial robustness. We argue that the deterministic approach enhances the robustness of VAEs when the latent codes are properly regularized. Consequently, we extend the training objective of the multi-modal deterministic autoencoders [27] to incorporate adversarially perturbed input data points in the latent space. The proposed adversarial learning scheme is comparatively less expensive and easier to implement than existing adversarially trained robust VAE models. We conduct extensive experimental analysis to evaluate the robustness of the trained model on popular benchmarks such as MNIST, FASHIONMNIST, SVHN and CELEBA images. Our empirical evaluations show that the proposed model consistently exhibits high adversarial robustness and significantly better generation performance compared to state-of-the-art robust VAE baseline models. We also show that by improving the robustness of the learned representations, a classifier trained on the learned latent space of the model also exhibits better robustness.

## 2 Related Work

The ability to defend against adversarial attacks is closely related to the sensitivity of the learned latent representations to slight changes in the input data points. In this section we first discuss seminal works proposed to generate rich and meaningful latent codes. This is followed by the review on adversaries for VAEs and strategies proposed to defend against such adversarial attacks.

**Variational Autoencoders (VAEs)** In standard VAEs, the encoder and the decoder are jointly trained to maximize the evidence lower bound (ELBO) of the log likelihood of the model. The resulting training objective is a combination of a reconstruction loss and the Kullback–Leibler (KL) divergence between the learned latent representations and a chosen prior - usually a uni-modal Gaussian normal distribution. Higgins et.al [16] improved the learned representation in VAEs by encouraging disentanglement in the latent space and introduces $\beta$-VAE for disentangled factor learning. $\beta$-VAE modifies VAEs by introducing a hyperparameter to balance the latent regularization term with the reconstruction performance. Chen et.al [7] further decompose the ELBO term in VAEs to introduce total correlation (TC) regularization and propose $\beta$-TCVAE as a promising alternative to $\beta$-VAEs. Willets et.al. [32] show that addition of TC term to the VAE objective also improves the robustness of VAEs. Several methods were proposed to include complex and flexible priors to the training pipeline of VAEs to enhance the semantics of the learned latent representations [8, 35, 31]. Miao et.al. [23] introduces an approach to incorporate the inductive bias into VAEs without explicitly changing the prior by utilizing intermediary set of latent variables. Hierarchical VAEs [26, 3, 34, 21] extend the standard VAE framework by introducing a hierarchy of latent variables and offer superior modelling capabilities. Gosh et.al. [12] question the variational formulation of VAEs and introduce a simple and effective deterministic model without any prior assumptions followed by a post-hoc density estimation to approximate the learned posterior. A multimodal version to this approach, that utilizes the capacity of expressive multi-modal latent distributions to yield high quality generation, was proposed in [27].

**Adversarial attacks on VAEs** Adversarial attacks targeted towards autoencoders were first discussed in [13]. Common attacks on VAEs follow procedures similar to attacks against classifiers, i.e. they aim to maximize the network's loss. Usually, slight perturbations are added to the input images to make the reconstructions similar to a specific target image (targeted/supervised attack) or a completely different

image (untargeted/unsupervised) [30, 19] such as to maximize the reconstruction loss. In [32] Willets et.al. show that applying the TC regularization introduced in TC-VAEs to hierarchical VAEs yields robust representations. Although the resulting model improves adversarial robustness, the training complexity is high when compared to a VAE. Cemgil at.al [6] relate the robustness of VAEs to the smoothness of the encoding process. Similarly to Madry et.al. [22], they introduce a regularization scheme based on a selection mechanism in the latent space to generate additional data points to minimize the entropy regularized Wasserstein distance between latent representations. Following the same direction, Cemgil et.al. [5] argue that the lack of consistency between the encoding-decoding process cause the susceptibility to adversarial attacks in VAES. In contrast to the previous works, Camuto et.al. [4] provide a theoretical insight to the robustness of VAEs and introduce a novel criterion for robustness in VAEs. Barret et.al. [2] propose to constrain the Lipschitz constants for both encoder and decoder to ensure certifiable adversarial robustness of VAEs. Although these methods comparatively improve the adversarial robustness of VAEs, they are often accompanied by complex network architectures and expensive training procedures. In contrast, our approach adopts an inexpensive adversarial training scheme to the latent space of deterministic autoencoders by an elegant extension to the regularization proposed in [27] to ensure both robustness and fidelity.

## 3 Towards robust deterministic autoencoders

Deterministic Autoencoders [12, 27] offer a promising alternative to VAEs for learning meaningful representations of complex input spaces with high fidelity. Motivated by this fact, we aim to further explore the robustness of the learned representations of the resulting model. We are particularly interested in the multimodal prior setting in [27], since a flexible and expressive Gaussian mixture model (GMM) prior assumption facilitates encoding similar data points closer together while distancing dissimilar points far in the latent space - a behaviour that has also been found to be beneficial in learning robust classifier models [11]. Consequently, we propose to adopt the regularization technique proposed in [27] to regularize the learned latent representations of our model towards a predefined GMM prior. For a model to be inherently robust, slight perturbations in the input space should not result in substantial variations in the encoding space and the corresponding reconstructions. This could be also attributed to the smoothness of the learned encoder. Hence, it is essential to investigate the smoothness of the learned representations and the reconstructions of the model. In the following subsections, we resume the regularization loss from [27] for the sake of completeness and illustrate how it can be extended to adversarial training samples.

### 3.1 Regularization of the learned representations – Encoder smoothness

Following the idea from [27], we aim to regularize the $D$-dimensional latent representations $\{\mathbf{z}_1, ..., \mathbf{z}_N\}$ of input datapoints $\{\mathbf{x}_1, ..., \mathbf{x}_N\}$ learned by the encoder $g$ of the model towards a $k$-modal GMM prior. For $k \leq K$, let $\boldsymbol{\mu}_k$ and $\boldsymbol{\Sigma}_k$ be the mean and covariance matrix of the $k$-th mode in the model. Further, let $w_k > 0$ be the weight of the $k$-th mode, $\sum_{k=1}^{K} w_k = 1$. Inspired by the statistical Kolmogorov-Smirnov (KS) test, the authors propose to regularize the empirical cumulative distribution function (CDF) of the latent samples towards the CDF $\Phi(\mathbf{z})$ of the prior distribution,

$$\Phi(\mathbf{z}) = \int_{-\infty}^{z_1} \cdots \int_{-\infty}^{z_D} \sum_{k=1}^{K} w_k \frac{\exp -\frac{1}{2}(\mathbf{t} - \boldsymbol{\mu}_k)^\top \boldsymbol{\Sigma}_k^{-1}(\mathbf{t} - \boldsymbol{\mu}_k)}{\sqrt{2\pi}^d |\boldsymbol{\Sigma}_k|} \, dt_1 \ldots dt_D, \tag{1}$$

where $\mathbf{z} = (z_1, \ldots, z_D), \mathbf{t} = (t_1, \ldots, t_D)$.

While the original KS-test quantifies a metric between *joint* CDFs, the minimization of this KS-distance can be approximated by marginalization, i.e. by regularizing the empirical *marginal* CDFs

$$\bar{F}_d(z) = \frac{1}{n} \sum_{n=1}^{N} \mathbb{1}_{[\mathbf{z}_n]_d \leq z} \tag{2}$$

of the latent representations $\{\mathbf{z}_1, ..., \mathbf{z}_N\}$ in dimensions $d = 1, \ldots, D$ towards the marginal CDFs

$$F_{\mathrm{GMM},d}(z) = \sum_{k=1}^{K} p_k \Phi\left(\frac{z - [\boldsymbol{\mu}_k]_d}{[\boldsymbol{\Sigma}_k]_{d,d}}\right) = \sum_{i=k}^{K} p_k \Phi\left(\frac{z - \mu_{k,d}}{\sigma_{k,d}}\right), \tag{3}$$

of the GMM prior. Here, $[\cdot]_d$ indicates the $d$-th dimension of a vector, i.e. $[\boldsymbol{\mu}_k]_d$ is the $d$-th entry of the mean vector $\boldsymbol{\mu}_k = (\mu_{k,1}, \ldots, \mu_{k,D})$ of mixture component $k$, i.e. a scalar. Similarly, $[\cdot]_{d,d}$ indicates the entry in row and column $d$ of the matrix $\boldsymbol{\Sigma}_k = \mathrm{diag}(\sigma_{k,1}, \ldots, \sigma_{k,D})$. The respective loss is computed as

$$\mathcal{L}_{\mathrm{KS},k}(\mathbf{z}_{1,\ldots,N}) = \frac{1}{D} \sum_{d=1}^{D} \mathrm{MSE}_{n=1}^{N} \left( \bar{F}_d([\mathbf{z}_n]_d), F_{\mathrm{GMM},d}([\mathbf{z}_n]_d) \right). \tag{4}$$

An additional loss compensates for the ambiguities on the target distribution introduced by the marginalization, i.e.

$$\mathcal{L}_{\mathrm{CV},k}(\mathbf{z}_{1,\ldots,N}) = \frac{1}{D^2} \sum_{\ell,d=1}^{D} \left( [\bar{\boldsymbol{\Sigma}}]_{\ell,d} - [\boldsymbol{\Sigma}^{\mathrm{GMM}}]_{\ell,d} \right)^2 \tag{5}$$

with $\boldsymbol{\Sigma}^{\mathrm{GMM}} = \sum_{k=1}^{K} w_k \boldsymbol{\Sigma}_k + \sum_{k=1}^{K} w_k (\boldsymbol{\mu}_k - \bar{\boldsymbol{\mu}}) (\boldsymbol{\mu}_k - \bar{\boldsymbol{\mu}})^T$, $\bar{\boldsymbol{\mu}} = \frac{1}{k} \sum_{k=1}^{K} \boldsymbol{\mu}_k$ and $\bar{\boldsymbol{\Sigma}}$ being the empirical covariance matrix of the latent representations.

The total loss of the model in [27] is a combination of both regularization terms to enforce the latent representations of the encoded data to match a predefined multi-modal prior distribution and a reconstruction loss. Motivated by [11], for equidistantly chosen modes $\boldsymbol{\mu}_i$ as in Figure 1, we expect such a model to already provide not only an improved generation fidelity as observed in [27] but also a more robust behavior. We provide empirical evidence for this conjecture in an ablation study in Section 4. The GMM regularization implicitly clusters latent points such that similar points are close to one another while dissimilar points are distant.

To further improve the robustness of our model we employ adversarial training - a widely popular defense strategy utilized to learn robust deep networks [22, 33]. That is, during training we utilize adversarial samples $\mathbf{z}_n^{\mathrm{adv}}$ that are explicitly optimized to fall in an $\epsilon-$ ball around the original latent representations $\mathbf{z}_n$ for $n \leq N$ and to decode to damaged or semantically altered images. In [1] it was observed that such adversarial samples tend to explore the underrepresented regions in the latent space. In the following, we extend the losses in (4) and (5) to overcome this undesired behaviour. This approach allows for a cheaper yet very effective adversarial training, while preserving the reconstruction and generation ability of the original model from [27].

## 3.2 Adversarial training data augmentation – Improving Robustness

To generate adversarial inputs, we adapt the fast gradient sign method [33] to the latent space of the model. For a given $\epsilon > 0$ and datapoint $\mathbf{x}_n$, the objective of the attack is to find the corresponding adversary $\mathbf{x}_n^{\mathrm{adv}}$ that would introduce maximum distance in the encoding space. That is, $\mathbf{x}_n^{\mathrm{adv}} = \mathbf{x}_n + \delta_{\mathbf{x}_n}$, where $\delta_{\mathbf{x}_n}$ is the solution to the optimization problem

$$\arg \max_{\delta} \| g(\mathbf{x}_n + \delta) - g(\mathbf{x}_n) \|_2 \text{ s.t. } \|\delta\|_\infty \leq \epsilon, \tag{6}$$

where $g$ is the encoder of the model. To prevent adversarial samples from exploring unexplored regions of the latent space, we assume that the joint distribution of latent encodings $(\mathbf{z}_n, \mathbf{z}_n^{\mathrm{adv}})$ (here $\mathbf{z}_n = g(x_n)$ and $\mathbf{z}_n^{\mathrm{adv}} = g(x_n + \delta_{x_n})$) of data points and their adversarial samples $(\mathbf{x}_n, \mathbf{x}_n^{\mathrm{adv}})$ to follow the same multi-modal GMM prior (Figure 1). One possible straight forward extension of the approach in [27] would be to consider adversarial examples as a specific data augmentation and regularize $\mathbf{z}_{1,\ldots,N}$ and $\mathbf{z}_{1,\ldots,N}^{\mathrm{adv}}$ to the same GMM prior independently and ignoring cross-covariance between $\mathbf{z}_{1,\ldots,N}$ and $\mathbf{z}_{1,\ldots,N}^{\mathrm{adv}}$ (here the off-diagonal elements in the covariance matrix of the GMM prior, that is the last matrix mentioned in equation (8), are zero) . The corresponding losses in eqs. (4) and (5) take the form

$$\mathcal{L}_{\mathrm{KS},k}^{\mathrm{aug}}(\mathbf{z}_{1,\ldots,N}, \mathbf{z}_{1,\ldots,N}^{\mathrm{adv}}) = \frac{1}{2} \mathcal{L}_{\mathrm{KS},k}(\mathbf{z}_{1,\ldots,N}) + \frac{1}{2} \mathcal{L}_{\mathrm{KS},k}(\mathbf{z}_{1,\ldots,N}^{\mathrm{adv}}) \tag{7}$$

and

$$\mathcal{L}_{\mathrm{CV},k}^{\mathrm{aug}}(\mathbf{z}_{1,\ldots,N}, \mathbf{z}_{1,\ldots,N}^{\mathrm{adv}}) = \frac{1}{4D^2} \sum_{\ell,d=1}^{2D} \left( \begin{bmatrix} \bar{\boldsymbol{\Sigma}} & \bar{\boldsymbol{\Sigma}}^{\mathrm{cross}} \\ \bar{\boldsymbol{\Sigma}}^{\mathrm{cross}} & \bar{\boldsymbol{\Sigma}}^{\mathrm{adv}} \end{bmatrix}_{\ell,d} - \begin{bmatrix} \boldsymbol{\Sigma}^{\mathrm{GMM}} & 0 \\ 0 & \boldsymbol{\Sigma}^{\mathrm{GMM}} \end{bmatrix}_{\ell,d} \right)^2, \tag{8}$$

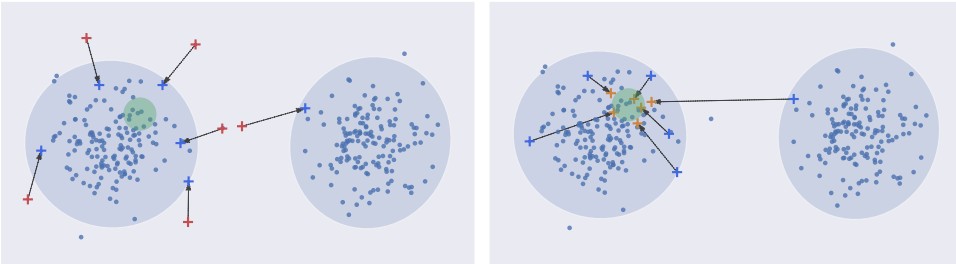

Figure 1: Left: Learned latent representations in a deterministic autoencoder regularized towards a GMM prior with two components (blue shaded regions). Consider a set of latent points $\mathbf{z}_1, \ldots, \mathbf{z}_N$ (blue dots), in a subspace (green shaded region) within a component and the corresponding adversarial examples $\mathbf{z}_1^{\mathrm{adv}}, \ldots, \mathbf{z}_N^{\mathrm{adv}}$ (red crosses). The adversarial examples tend to explore regions not covered by the input samples. If we assume that $\mathbf{z}$ and $\mathbf{z}^{\mathrm{adv}}$ follow the same prior assumptions independently, the adversarial examples tend to move closer to the original samples (blue crosses). In the worst case scenario, an adversarial example might reside in a different component. Right: By establishing a strong coupling via a 2-point KS-distance regularization, the adversarial examples tend to move more closer to the original samples (orange crosses) after regularization.

where $\bar{\boldsymbol{\Sigma}}$ and $\bar{\boldsymbol{\Sigma}}^{\mathrm{adv}}$ are the empirical covariance matrices of the latent representations $\mathbf{z}_{1,\ldots,N}$ and their adversaries $\mathbf{z}_{1,\ldots,N}^{\mathrm{adv}}$ respectively, and $\bar{\boldsymbol{\Sigma}}_{\mathrm{cross}}$ is the empirical cross-covariance between benign and adversarial samples. While such a regularization preserves the overall distribution even under adversarial attacks, it can not control the distance of a specific adversarial sample to its benign $\mathbf{z}_n$. In the worst case scenario, an adversarial example $\mathbf{z}_n^{\mathrm{adv}}$ can be mapped to a different Gaussian mixture component than $\mathbf{z}_n$ and therefore cause maximum damage in the reconstruction as shown in Figure 1(left).

To evenly spread out the learned representations, we inject Gaussian noise to the latent vectors during training. Let $\mathbf{x}_{\epsilon,n}$ be the output of the decoder at $\mathbf{z}_n + \epsilon_n$, where $\epsilon_n \sim \mathcal{N}(0, I_D)$. The reconstruction loss equals the mean squared error between inputs $\mathbf{x}_n$ and their noisy reconstructions $\mathbf{x}_{\epsilon,n}$.

### 3.3 Coupling of $\mathbf{z}$ and $\mathbf{z}^{\mathrm{adv}}$ – A Two-Point KS-distance loss

To ensure that the adversarial examples remain in close proximity to the original mapping in the learned latent space, we establish a strong coupling between the two distributions, $\mathbf{z}_{1,\ldots,N}$ and $\mathbf{z}_{1,\ldots,N}^{\mathrm{adv}}$. Hence we propose to match the the empirical CDFs of $\mathbf{z}_{1,\ldots,N}$ and $\mathbf{z}_{1,\ldots,N}^{\mathrm{adv}}$ and introduce a novel regularization based on the *two-point KS-test* [29]. By analogy to the one-point KS-test that tests whether a sample is drawn from a given, continuous distribution, the two-sample KS test determines whether two samples with empirical CDF are drawn from the same distribution. To this end, the two-sample KS test evaluates the supremum of the distance between the two CDFs. Here, we propose to minimize this distance computed from the marginalized two-point KS-test in order to align the distributions of benign points and their adversaries. The resulting loss is consistent with the previous regularization and allows to efficiently establish the desired coupling between the representations. The first regularization loss of the adversarially extended model with pairwise coupling takes the following form

$$\mathcal{L}_{\mathrm{KS},k}^{\mathrm{adv}}(\mathbf{z}_{1,\ldots,N}, \mathbf{z}_{1,\ldots,N}^{\mathrm{adv}}) = \frac{2}{3}\mathcal{L}_{\mathrm{KS},k}^{\mathrm{aug}}(\mathbf{z}_{1,\ldots,N}, \mathbf{z}_{1,\ldots,N}^{\mathrm{adv}}) + \frac{1}{3D}\sum_{d=1}^{D}\mathrm{MSE}_{n=1}^{N}\left(\bar{F}_d([\mathbf{z}_n]_d), \bar{F}_d^{\mathrm{adv}}([\mathbf{z}_n^{\mathrm{adv}}]_d)\right)$$

(9)

where $\bar{F}_d, \bar{F}_d^{\mathrm{adv}}$ are the empirical CDFs of $\mathbf{z}$ and $\mathbf{z}^{\mathrm{adv}}$ respectively.

The correlations between the latent representations and their adversarial samples need to be considered separately. The degree or strength of the coupling is controlled by a coupling parameter $|\alpha| \leq 1$, where $\alpha = 1$ indicates the border condition where $\mathbf{z} = \mathbf{z}^{\mathrm{adv}}$. The covariance loss of the extended

model becomes,

$$\mathcal{L}_{\mathrm{CV},k}^{\mathrm{adv}}(\mathbf{z}_{1,\ldots,N}, \mathbf{z}_{1,\ldots,N}^{\mathrm{adv}}) = \frac{1}{4D^2} \sum_{\ell,d=1}^{2D} \left( \begin{bmatrix} \bar{\boldsymbol{\Sigma}} & \bar{\boldsymbol{\Sigma}}^{\mathrm{cross}} \\ \bar{\boldsymbol{\Sigma}}^{\mathrm{cross}} & \bar{\boldsymbol{\Sigma}}^{\mathrm{adv}} \end{bmatrix}_{\ell,d} - \begin{bmatrix} \boldsymbol{\Sigma}^{\mathrm{GMM}} & \alpha \boldsymbol{\Sigma}^{\mathrm{GMM}} \\ \alpha \boldsymbol{\Sigma}^{\mathrm{GMM}} & \boldsymbol{\Sigma}^{\mathrm{GMM}} \end{bmatrix}_{\ell,d} \right)^2 . \tag{10}$$

The total training objective of the model takes the following form,

$$\mathcal{L}(\mathbf{x}_{1,\ldots,N}, \mathbf{x}_{1,\ldots,N}^{\mathrm{adv}}) = \lambda_{\mathrm{REC}} \mathcal{L}_{\mathrm{REC}}(\mathbf{x}_{1,\ldots,N}^{\epsilon}) + \lambda_{\mathrm{KS}} \mathcal{L}_{\mathrm{KS},k}^{\mathrm{adv}}(\mathbf{z}_{1,\ldots,N}, \mathbf{z}_{1,\ldots,N}^{\mathrm{adv}}) +$$
$$\lambda_{\mathrm{CV}} \mathcal{L}_{\mathrm{CV},k}^{\mathrm{adv}}(\mathbf{z}_{1,\ldots,N}, \mathbf{z}_{1,\ldots,N}^{\mathrm{adv}}). \tag{11}$$

The weights $\lambda_{\mathrm{REC}}$, $\lambda_{\mathrm{KS}}$ and $\lambda_{\mathrm{CV}}$ can be calculated by taking the statistics of samples from the GMM prior [27], see Appendix.

## 4 Experiments

We conduct an extensive experimental analysis to evaluate the robustness of the proposed model. We consider the state of the art latent space attacks targeted at VAEs [13, 19, 2] and evaluate the robustness based on the quantitative metrics as described in Section 4. Since the latent spaces of VAEs are often further utilized for various downstream applications, we also consider the impact of such adversarial attacks on a classifier trained in the latent space of the model. To evaluate the fidelity of the learned representations, we report the Fréchet inception distance (FID) [15] of the generated images. Our model is compared with the following baseline models, Variational Autoencoder (VAE) [18], $\beta$-VAE [16], $\beta$-TCVAE [7], LipschitzVAE [2], Smooth Encoders (SE) [6] and Autoencoding Variational Autoencoder (AVAE) [5]. The experimental study is conducted on important image benchmark datasets such as MNIST, FASHIONMNIST, SVHN and CELEBA. For simplicity we consider a fully connected network architecture for experiments on MNIST and FASHIONMNIST images and a convolutional architecture for experiments on SVHN and CELEBA images. Further details of the experiments are given in the Appendix[1].

**Adversarial attacks.** Adversarial attacks targeted at VAEs attempt to add small noise perturbations to the input data points that fool the model to reconstruct the input image to a target adversarial image or a completely different image. Following recent literature [2, 32], we consider two types of adversarial attacks in our experiments.

Latent space attacks or supervised attacks are considered the most effective mode of attacks on VAEs. Here, the attacker tries to add a noise perturbation $\delta$ to a data point $\mathbf{x}$, such that the latent representation $\mathbf{z}_{x+\delta}$ of the perturbed input $\mathbf{x} + \delta$ is close to the latent representation $\mathbf{z}_t$ of a chosen target adversarial image, $\mathbf{x}_t$. The attack involves solving the following optimization problem,

$$\arg \min_{\|\delta\|_2 \le \lambda} \|(\mathbf{z}_{x+\delta} - \mathbf{z}_t)\|_2. \tag{12}$$

Further, we consider maximum damage or output space attack. In this setting, the adversary perturbs the input data point to cause maximum damage in the reconstruction of the decoder $f$ of the model and optimizes the following objective,

$$\arg \max_{\|\delta\|_2 \le \lambda} \|f(\mathbf{z}_{x+\delta}) - f(\mathbf{z}_x)\|_2. \tag{13}$$

In both scenarios, the noise perturbation is explicitly constrained by some constant $\lambda > 0$ to ensure a consistent comparison with the baseline models.

**Evaluation.** The qualitative results for CELEBA images for latent space and maximum damage attack are shown in Figure 3. We consider three evaluation metrics to quantitatively estimate the robustness of the different models for the above-mentioned adversarial attacks. First, we evaluate the *attack loss* of the adversary. That is, we report the achieved value of the optimization objectives in (12) and (13). The observed attack losses for the two forms of adversarial attacks are shown in Figure 2 (MNIST results are given in the Appendix). Higher attack losses correspond to less

---

[1]All the conducted experiments were part of a carbon-neutral framework based GPU cluster and hence did not contribute to climate change.

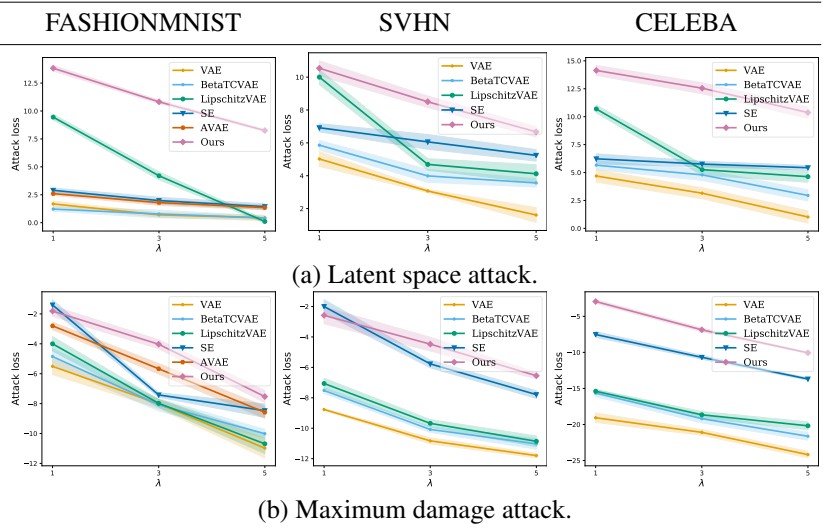

(a) Latent space attack.

(b) Maximum damage attack.

Figure 2: Observed attack losses for (a) latent space attack (eqn (12)) and (b) maximum damage attack (eqn (13)) with varying $\lambda$ values. We report the observed mean and standard deviation by attacking 100 randomly chosen test images in 10 different trials. Higher loss indicates more robustness.

successful attacks and hence better robustness. Due to the different regularization methods used in the baseline models, the inherent scale of the aggregated posterior changes and hence the value of the attack objective in eqn 12. Hence for latent space attacks, the values reported in Figure 2 might not be directly comparable. Another strong indicator for the robustness of the considered models is the similarity between the reference image $\mathbf{x}_r$ and the reconstructions $\tilde{\mathbf{x}}_a$ of the adversarially perturbed variants $\mathbf{x}_a$. In the case of latent space attacks, we consider the target image $\mathbf{x}_r = \mathbf{x}_t$ as the reference images and compare with the reconstruction $\tilde{\mathbf{x}}_a = f(\mathbf{z}_{\mathbf{x}+\delta})$ of the perturbed latent representation. For a maximum damage attack, we consider the original images $\mathbf{x}_r = \mathbf{x}$ as the reference images and compare them to $\tilde{\mathbf{x}}_a$. Similar to [20], we use the perception-based similarity metric Multi-Scale Structural Similarity Index Measure (MSSSIM $\in [0, 1]$ ) to compare the images. We also report the $l_2$-distance as an alternative similarity metric to MSSSIM in the Appendix. Lower values of $\text{MSSSIM}(x_r, \tilde{x}_a)$ indicate less similarity between the reference (target) image and the adversarial reconstructions and correspond to less successful latent space attacks. And higher value of $\text{MSSSIM}(\tilde{x}_r, \tilde{x}_a)$ corresponds to less successful maximum damage attacks as they correspond to high similarity between reference (original) and the adversarial reconstructions. We report the observed values in Table 1. Finally, it is also essential to evaluate how similar the adversarial images are to the original image. Ideally, a successful attack implies that both adversarial and original images look similar in appearance (MSSSIM $\approx 1$). Hence we consider the MSSSIM between the original and adversarial images in Table 1. We also report the FID of the generated samples to compare the fidelity of the representations.

**Results.** Overall we see that the proposed model outperforms all the considered baselines in terms of both robustness and offers superior generation performance. Even for complex datasets like SVHN and CELEBA, we observe the same trend with FID and robustness measures. These results are especially promising since we did not employ any extensive hyperparameter search for training. Our results further confirm that both robust and high fidelity models are possible. Since we employ an FGSM based adversarial training, the training time required is cheaper when compared to the expensive PGD-based training used in smooth encoders (SE). The computation time for a single iteration of SEs is two times more when compared to our method.

**Ablation Study.** In this section, we compare three different variants of regularized deterministic autoencoders to evaluate the importance of joint regularization of the original and adversarial samples. We begin with the model proposed in [27], which we denote as GMM-DAE. Second, we study the augmented model defined by equations (7) and (8) in Section 3.2, but without the coupling of original and adversarial latent representations (Augmented). We compare the robustness and fidelity of these models with our proposed model, i.e. where original and adversarial latent representations are not

Table 1: Robustness evaluation across dataset - similarities between images in the event of latent space and maximum damage attacks in terms of MSSSIM. Here randomly chosen 100 test images are attacked in 10 different trials. $x_r$ refers to reference image, $x_a$ to adversarial image and $\tilde{x}_r$, $\tilde{x}_a$ to their corresponding reconstructions. The maximum input noise perturbation levels $\lambda$ are limited to $1, 3$ and $5$. Fidelity analysis - based on the FID of the generated images.

| MNIST | Latent space attack | | | | | | Maximum damage attack | | | | | | FID(↓) |
|---|---|---|---|---|---|---|---|---|---|---|---|---|---|
| | MSSSIM($\mathbf{x}_r, \tilde{\mathbf{x}_a}$)(↓) | | | MSSSIM($\mathbf{x}_r, \mathbf{x}_a$)(↓) | | | MSSSIM($\tilde{\mathbf{x}_r}, \tilde{\mathbf{x}_a}$)(↑) | | | MSSSIM($\mathbf{x}_r, \mathbf{x}_a$)(↓) | | | |
| | 1 | 3 | 5 | 1 | 3 | 5 | 1 | 3 | 5 | 1 | 3 | 5 | |
| VAE | 0.55 | 0.78 | 0.89 | 0.99 | 0.94 | 0.87 | 0.64 | 0.27 | 0.08 | 0.98 | 0.93 | 0.86 | 43.21 |
| β-VAE | 0.52 | 0.73 | 0.86 | 0.99 | 0.92 | 0.86 | 0.65 | 0.36 | 0.22 | 0.98 | 0.93 | 0.87 | 42.72 |
| β-TCVAE | 0.53 | 0.69 | 0.83 | 0.98 | 0.92 | 0.86 | 0.73 | 0.38 | 0.28 | 0.98 | 0.93 | 0.87 | 45.61 |
| LipschitzVAE | 0.50 | 0.68 | 0.79 | 0.98 | 0.93 | 0.89 | 0.75 | 0.41 | 0.33 | 0.98 | 0.93 | 0.86 | 59.45 |
| SE | 0.49 | 0.62 | 0.68 | 0.98 | 0.92 | 0.86 | 0.90 | 0.60 | 0.54 | 0.98 | 0.93 | 0.86 | 47.34 |
| AVAE | 0.49 | 0.59 | 0.62 | 0.98 | 0.91 | 0.83 | 0.80 | 0.65 | 0.59 | **0.97** | 0.89 | 0.86 | 48.47 |
| Ours | **0.38** | **0.47** | **0.60** | **0.95** | **0.90** | **0.80** | **0.92** | **0.82** | **0.69** | 0.98 | **0.89** | **0.78** | **39.37** |

| FASHIONMNIST | Latent space attack | | | | | | Maximum damage attack | | | | | | FID(↓) |
|---|---|---|---|---|---|---|---|---|---|---|---|---|---|
| | MSSSIM($\mathbf{x}_r, \tilde{\mathbf{x}_a}$)(↓) | | | MSSSIM($\mathbf{x}_r, \mathbf{x}_a$)(↓) | | | MSSSIM($\tilde{\mathbf{x}_r}, \tilde{\mathbf{x}_a}$)(↑) | | | MSSSIM($\mathbf{x}_r, \mathbf{x}_a$)(↓) | | | |
| | 1 | 3 | 5 | 1 | 3 | 5 | 1 | 3 | 5 | 1 | 3 | 5 | |
| VAE | 0.61 | 0.65 | 0.71 | 0.98 | 0.91 | 0.82 | 0.58 | 0.29 | 0.13 | 0.99 | 0.94 | 0.87 | 70.22 |
| β-VAE | 0.59 | 0.61 | 0.68 | 0.98 | 0.92 | 0.82 | 0.66 | 0.32 | 0.15 | 0.99 | 0.94 | 0.85 | 73.82 |
| β-TCVAE | 0.55 | 0.58 | 0.64 | 0.98 | 0.92 | 0.82 | 0.69 | 0.35 | 0.27 | 0.99 | 0.94 | 0.87 | 73.94 |
| LipschitzVAE | 0.43 | 0.59 | 0.67 | 0.99 | 0.94 | 0.89 | 0.71 | 0.34 | 0.30 | 0.99 | 0.94 | 0.88 | 79.45 |
| SE | 0.24 | 0.43 | 0.53 | 0.98 | 0.92 | **0.81** | 0.90 | 0.62 | 0.43 | 0.99 | 0.94 | 0.86 | 72.29 |
| AVAE | 0.32 | 0.34 | **0.35** | 0.98 | 0.92 | 0.82 | 0.79 | 0.52 | 0.45 | 0.99 | 0.94 | 0.92 | 74.45 |
| Ours | **0.22** | **0.26** | 0.39 | **0.97** | **0.91** | 0.81 | **0.92** | **0.77** | **0.63** | **0.97** | **0.92** | **0.83** | **64.89** |

| SVHN | Latent space attack | | | | | | Maximum damage attack | | | | | | FID(↓) |
|---|---|---|---|---|---|---|---|---|---|---|---|---|---|
| | MSSSIM($\mathbf{x}_r, \tilde{\mathbf{x}_a}$)(↓) | | | MSSSIM($\mathbf{x}_r, \mathbf{x}_a$)(↓) | | | MSSSIM($\tilde{\mathbf{x}_r}, \tilde{\mathbf{x}_a}$)(↑) | | | MSSSIM($\mathbf{x}_r, \mathbf{x}_a$)(↓) | | | |
| | 1 | 3 | 5 | 1 | 3 | 5 | 1 | 3 | 5 | 1 | 3 | 5 | |
| VAE | 0.46 | 0.76 | 0.87 | 0.98 | 0.87 | 0.77 | 0.55 | 0.46 | 0.38 | 0.99 | 0.93 | 0.89 | 58.98 |
| β-VAE | 0.44 | 0.70 | 0.81 | 0.99 | 0.89 | 0.77 | 0.52 | 0.49 | 0.47 | 0.99 | 0.93 | 0.88 | 61.65 |
| β-TCVAE | 0.39 | 0.65 | 0.72 | 0.99 | 0.89 | 0.77 | 0.63 | 0.60 | 0.54 | 0.99 | **0.92** | 0.88 | 62.59 |
| LipschitzVAE | 0.35 | 0.62 | 0.71 | 0.99 | 0.89 | **0.76** | 0.66 | 0.65 | 0.55 | 0.99 | 0.93 | 0.88 | 65.58 |
| SE | 0.19 | 0.33 | 0.34 | 0.99 | 0.92 | 0.81 | 0.79 | 0.69 | 0.60 | 0.99 | 0.96 | 0.94 | 61.28 |
| Ours | **0.16** | **0.26** | **0.28** | **0.98** | **0.77** | 0.76 | **0.84** | **0.79** | **0.75** | 0.98 | 0.92 | 0.86 | **38.89** |

| CELEBA | Latent space attack | | | | | | Maximum damage attack | | | | | | FID(↓) |
|---|---|---|---|---|---|---|---|---|---|---|---|---|---|
| | MSSSIM($\mathbf{x}_r, \tilde{\mathbf{x}_a}$)(↓) | | | MSSSIM($\mathbf{x}_r, \mathbf{x}_a$)(↓) | | | MSSSIM($\tilde{\mathbf{x}_r}, \tilde{\mathbf{x}_a}$)(↑) | | | MSSSIM($\mathbf{x}_r, \mathbf{x}_a$)(↓) | | | |
| | 1 | 3 | 5 | 1 | 3 | 5 | 1 | 3 | 5 | 1 | 3 | 5 | |
| VAE | 0.59 | 0.60 | 0.66 | 0.99 | 0.99 | 0.97 | 0.64 | 0.58 | 0.55 | 0.99 | 0.98 | 0.98 | 69.48 |
| β-VAE | 0.55 | 0.58 | 0.64 | 0.99 | 0.99 | 0.97 | 0.68 | 0.60 | 0.59 | 0.99 | 0.98 | 0.97 | 75.65 |
| β-TCVAE | 0.54 | 0.51 | 0.61 | 0.99 | 0.99 | 0.97 | 0.76 | 0.71 | 0.64 | 0.98 | 0.99 | **0.96** | 75.11 |
| LipschitzVAE | 0.49 | 0.51 | 0.55 | 0.99 | **0.98** | 0.97 | 0.73 | 0.70 | 0.64 | 0.98 | **0.98** | **0.96** | 77.89 |
| SE | **0.27** | 0.31 | 0.34 | 0.99 | **0.98** | 0.96 | **0.97** | 0.91 | 0.76 | 0.99 | **0.98** | 0.98 | 72.68 |
| Ours | 0.28 | **0.29** | **0.32** | 0.99 | **0.98** | 0.96 | **0.97** | **0.93** | **0.80** | 0.99 | **0.98** | 0.96 | **51.98** |

Table 2: Ablation study on MNIST images. Augmented refers to the model definition in eqs 7, 8. Here $x_r$ refers to reference image, $x_a$ to adversarial image and $\tilde{x}_r$, $\tilde{x}_a$ to their corresponding reconstructions. The maximum input noise perturbation level $\lambda$ is limited to $1, 3$ and $5$.

| Method | Latent space attack | | | Maximum damage attack | | | FID(↓) |
|---|---|---|---|---|---|---|---|
| | MSSSIM($x_r, \tilde{x}_a$)(↓) | | | MSSSIM($\tilde{x}_r, \tilde{x}_a$)(↑) | | | |
| | 1 | 3 | 5 | 1 | 3 | 5 | |
| GMM-DAE [27] | 0.54 | 0.70 | 0.82 | 0.75 | 0.37 | 0.30 | **38.89** |
| Augmented | 0.47 | 0.59 | 0.71 | 0.79 | 0.56 | 0.54 | 40.16 |
| Ours | **0.38** | **0.47** | **0.60** | **0.92** | **0.82** | **0.69** | 39.37 |

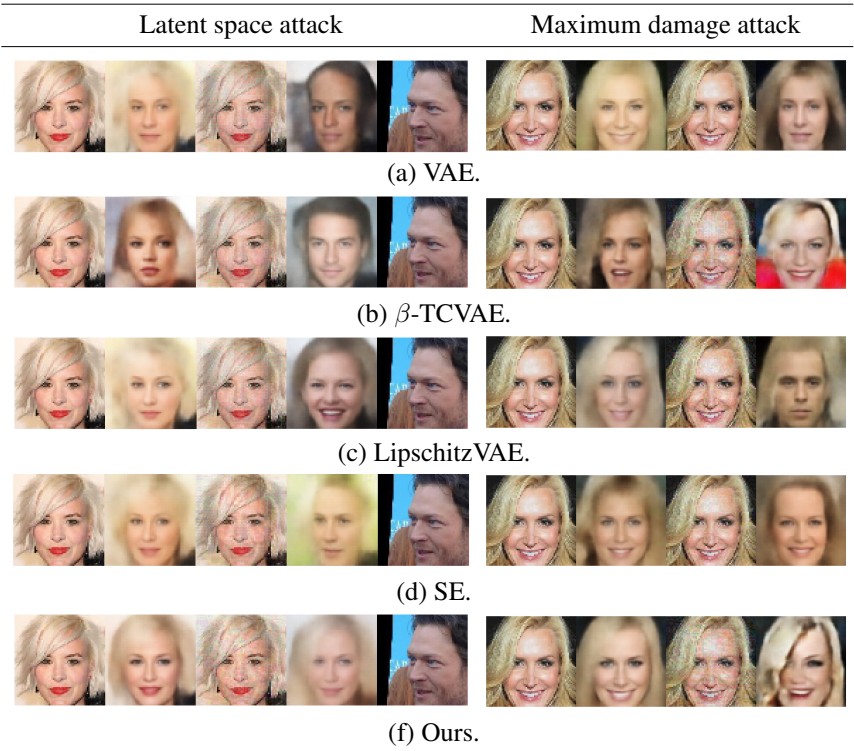

| Latent space attack | Maximum damage attack |

(a) VAE.

(b) $\beta$-TCVAE.

(c) LipschitzVAE.

(d) SE.

(f) Ours.

Figure 3: Visual appraisal of attacks - Qualitative analysis across models on CELEBA images with maximum input noise perturbation level $\lambda$ limited to 5 for latent space and maximum damage attack. (from left to right) Images in each row correspond to input image($x$), clean reconstruction, adversarial image($x_a$), adversarial reconstruction($\tilde{\mathbf{x}}_a$) and target image($x_t$) (for latent space attacks) respectively.

Table 3: Robustness of downstream classifier trained in the latent space of the model under adversarial attack - we report the clean accuracy and the accuracy during attack defined in eqn 12, for $\lambda = 1$.

| Method | MNIST | | FASHIONMNIST | | SVHN | |
|---|---|---|---|---|---|---|
| | clean acc.($\uparrow$) | $\lambda = 1(\uparrow)$ | clean acc.($\uparrow$) | $\lambda = 1(\uparrow)$ | clean acc.($\uparrow$) | $\lambda = 1(\uparrow)$ |
| VAE | 92.16 | 58.85 | 80.65 | 59.15 | 61.70 | 25.63 |
| $\beta$-TCVAE | 93.02 | 61.06 | 81.25 | 60.77 | 62.02 | 30.12 |
| LipschitzVAE | 90.78 | 62.00 | 80.50 | 62.06 | 60.99 | 33.99 |
| SE | 93.81 | 68.65 | 80.10 | 66.83 | 62.36 | 44.40 |
| Ours | **96.08** | **91.78** | **85.96** | **78.86** | **70.96** | **59.20** |

only regularized towards the same prior but coupled according to equations (9), (10), and (11) in Section 3.3 (Ours). The observed metrics are reported in Table 2. We observe that the proposed method yields comparatively better performance in terms of robustness while still maintaining the generation fidelity when compared to the non-robust version, GMM-DAE. It can also be seen that enforcing coupling between the latent representations of the original and adversarial samples (Ours) leads to better performance than simply augmenting them (Augmented). It is worth pointing out that the GMM-DAE maintains better performance than a standard VAE model (Table 1, MNIST VAE results). This is due to the well-structured latent space in GMM-DAE. This observation further confirms our hypothesis in Section 3 that when similar-looking samples are modelled together in the latent space of the model, robustness can be improved.

**Robustness to downstream applications.** Since the learned representations of VAEs are often used for various downstream tasks, it is also vital to verify how adversarial attacks affect the performance of the same. To showcase the effect of adversarial attacks to downstream classification tasks, we

train a MLP classifier on the latent space of the model and observe the accuracy drop in the presence of the latent space attacks for a constrained noise norm $\lambda = 1$. The observed values along with the clean accuracy are shown in Table 3. The classifier trained on the latent space of the proposed model achieves better accuracies when compared to the baseline models under latent space attack.

## 5    Discussion & Conclusion

**Limitations.**  Although cheaper when compared to the current adversarially trained robust VAE models, the FGSM based adversarial training scheme is still expensive when compared to the non-robust counterpart. The coupling parameter in the proposed method is an additional hyperparameter to tune and enforcing strong coupling i.e. $\alpha = 1$, might compromise the generation fidelity of the resulting model. While we need the coupling to perform adversarial training, overly strong coupling will necessarily lead to deteriorated reconstructions, similar as in VAEs. It is worth noting that we did not perform intensive hyperparameter optimization in this line of experiments. Hyperparameter optimization might be needed in other datasets for this reason. Further, datasets with large inter class variance might not benefit from the underlying multimodal prior assumptions in the current approach. It would be interesting to explore the impact of the adversarial attacks in other potential downstream applications such as optimization in the latent space of VAE models.

**Potential societal impact.**  Variational autoencoders enable learning meaningful representations of complex high dimensional data without any supervision. This further enhances the usability of these learned representations for various downstream tasks and applications where limited data is available for example, due to privacy/security concerns. Hence, it is important to study the robustness of these representations along with their accuracy, especially when employed in real-world applications. In the present paper, we propose a method to train robust VAEs with high fidelity in the learned latent space. Since our work is a step towards more robust models, we hope to see a positive social impact. There is currently limited work in this direction and we believe that our method encourages potential future work in developing robust VAE models. On the other hand, we should take into consideration the possible negative social impact of this research, especially in safety-critical applications. Although we observe superior robustness in our model against the existing attacks, similar performance cannot be guaranteed on newly discovered attacks on VAEs. Hence, when deployed in real-world applications we highly recommend testing the model continuously against newly designed attacks. We also urge the machine learning community to pursue this work responsibly to enable potential future research without any misuse.

**Conclusion.**  Developing robust VAE models is crucial since the learned representations of VAEs are frequently used for various applications. Motivated by the recent research towards deterministic alternatives to VAEs, we study the robustness of deterministic autoencoders. We extend recently developed regularization schemes to efficiently couple the adversarial examples and the learned representations during training. Our experiments show that with proper regularization, adversarially trained multimodal deterministic autoencoders offer significantly improved adversarial robustness and high fidelity in the learned latent space.

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
