# Trading off Image Quality for Robustness is not Necessary with Deterministic Autoencoders

**Amrutha Saseendran**[1], **Kathrin Skubch**[1], **Stefan Falkner**[1], **and Margret Keuper**[2,3]
[1]Bosch Center for Artificial Intelligence
[2]University of Siegen, [3]Max Planck Institute for Informatics, Saarland Informatics Campus
`Amrutha.Saseendran@de.bosch.com`

## A    Appendix

In this section, we provide additional results to support the main paper. The document is structured as follows.

### A.1    Unimodal robust deterministic autoencoders

The objective of the model is to regularize $D$-dimensional latent representations $\{\mathbf{z}_1, ..., \mathbf{z}_N\}$ of input datapoints $\{\mathbf{x}_1, ..., \mathbf{x}_N\}$ learned by the encoder $g$ of the model towards a unimodal Gaussian $Z \sim \mathcal{N}(\mu, \mathbf{\Sigma})$ with mean $\boldsymbol{\mu}$ and covariance matrix $\mathbf{\Sigma}$. Inspired by the statistical Kolmogorov-Smirnov (KS) test, the empirical cumulative distribution function (CDF) of the latent samples is regularized towards the CDF $\Phi(\mathbf{z})$ of the prior distribution  [13],

$$\Phi(\mathbf{z}) = \int_{-\infty}^{z_1} \cdots \int_{-\infty}^{z_D} \frac{\exp -\frac{1}{2}(\mathbf{t} - \boldsymbol{\mu})^\top \mathbf{\Sigma}^{-1}(\mathbf{t} - \boldsymbol{\mu})}{\sqrt{2\pi}^d |\mathbf{\Sigma}|} \, dt_1 \ldots dt_D, \tag{1}$$

where $\mathbf{z} = (z_1, \ldots, z_D), \mathbf{t} = (t_1, \ldots, t_D)$.

To avoid the computational complexity of considering in the original KS-distance formulation that requires computing *joint* CDFs, the minimization of this KS-distance can be approximated by

marginalization, i.e. by regularizing the empirical *marginal* CDFs

$$\bar{F}_d(z) = \frac{1}{n} \sum_{n=1}^{N} \mathbb{1}_{[\mathbf{z}_n]_d \leq z} \tag{2}$$

of the latent representations $\{\mathbf{z}_1, ..., \mathbf{z}_N\}$ in dimensions $d = 1, \ldots, D$ towards the marginal CDFs

$$F_{\mathcal{N},d}(z) = \Phi\left(\frac{z - [\boldsymbol{\mu}]_d}{[\boldsymbol{\Sigma}]_{d,d}}\right) = \Phi\left(\frac{z - \mu_d}{\sigma_d}\right), \tag{3}$$

of the Gaussian prior. Here, $[\cdot]_d$ indicates the $d$-th dimension of a vector, i.e. $[\boldsymbol{\mu}]_d$ is the $d$-th entry of the mean vector $\boldsymbol{\mu} = (\mu_1, \ldots, \mu_D)$. Similarly, $[\cdot]_{d,d}$ indicates the entry in row and column $d$ of the matrix $\boldsymbol{\Sigma}_k = \text{diag}(\sigma_1, \ldots, \sigma_D)$. The respective loss is computed as

$$\mathcal{L}_{\text{KS},k}(\mathbf{z}_{1,\ldots,N}) = \frac{1}{D} \sum_{d=1}^{D} \underset{n=1}{\overset{N}{\text{MSE}}} \left( \bar{F}_d([\mathbf{z}_n]_d), F_{\mathcal{N},d}([\mathbf{z}_n]_d) \right). \tag{4}$$

To account for the correlations between the different latent dimensions, an additional covariance based loss is defined as follows,

$$\mathcal{L}_{\text{CV},k}(\mathbf{z}_{1,\ldots,N}) = \frac{1}{D^2} \sum_{\ell,d=1}^{D} \left( [\bar{\boldsymbol{\Sigma}}]_{\ell,d} - [\boldsymbol{\Sigma}]_{\ell,d} \right)^2 \tag{5}$$

where $\bar{\boldsymbol{\Sigma}}$ is the empirical covariance matrix of the latent representations and $\boldsymbol{\Sigma}$ stands for the prior covariance. The total loss of the model is a combination of both regularization terms and a reconstruction loss.

To improve the robustness of the model we utilize adversarial training [11, 15] similar to the main paper. That is, during training we augment the input data points with adversarial samples whose latents $\mathbf{z}_n^{\text{adv}}$ are explicitly optimized to fall in an $\epsilon-$ ball around the original latent representations $\mathbf{z}_n$ for $n \leq N$ and to decode to semantically altered images. To ensure that the adversarial examples remain in close proximity to the original mapping in the learned latent space, we establish a strong coupling between the two distributions by introducing *two-point KS-test* [14] loss.

**Two-Point KS-distance loss for unimodal prior**   The first regularization loss of the adversarially extended model with pairwise coupling takes the following form,

$$\mathcal{L}_{\text{KS},k}^{\text{adv}}(\mathbf{z}_{1,\ldots,N}, \mathbf{z}_{1,\ldots,N}^{\text{adv}}) = \frac{2}{3} \mathcal{L}_{\text{KS},k}^{\text{aug}}(\mathbf{z}_{1,\ldots,N}, \mathbf{z}_{1,\ldots,N}^{\text{adv}}) + \frac{1}{3D} \sum_{d=1}^{D} \underset{n=1}{\overset{N}{\text{MSE}}} \left( \bar{F}_d([\mathbf{z}_n]_d), \bar{F}_d^{\text{adv}}([\mathbf{z}_n^{\text{adv}}]_d) \right) \tag{6}$$

where $\bar{F}_d, \bar{F}_d^{\text{adv}}$ are the empirical CDFs of $\mathbf{z}$ and $\mathbf{z}^{\text{adv}}$ respectively.

The correlations between the latent representations and their adversarial samples need to be considered separately as before. The covariance loss of the extended model is defined as follows,

$$\mathcal{L}_{\text{CV},k}^{\text{adv}}(\mathbf{z}_{1,\ldots,N}, \mathbf{z}_{1,\ldots,N}^{\text{adv}}) = \frac{1}{2D^2} \sum_{\ell,d=1}^{2D} \left( \begin{bmatrix} \bar{\boldsymbol{\Sigma}} & \bar{\boldsymbol{\Sigma}}^{\text{cross}} \\ \bar{\boldsymbol{\Sigma}}^{\text{cross}} & \bar{\boldsymbol{\Sigma}}^{\text{adv}} \end{bmatrix}_{\ell,d} - \begin{bmatrix} \boldsymbol{\Sigma} & \alpha\boldsymbol{\Sigma} \\ \alpha\boldsymbol{\Sigma} & \boldsymbol{\Sigma} \end{bmatrix}_{\ell,d} \right)^2. \tag{7}$$

where $\alpha \leq 1$ is the coupling parameter. Since we consider a Gaussian prior with zero mean and identity covariance, $Z \sim \mathcal{N}(0, I)$, the covariance loss becomes,

$$\mathcal{L}_{\text{CV},k}^{\text{adv}}(\mathbf{z}_{1,\ldots,N}, \mathbf{z}_{1,\ldots,N}^{\text{adv}}) = \frac{1}{2D^2} \sum_{\ell,d=1}^{2D} \left( \begin{bmatrix} \bar{\boldsymbol{\Sigma}} & \bar{\boldsymbol{\Sigma}}^{\text{cross}} \\ \bar{\boldsymbol{\Sigma}}^{\text{cross}} & \bar{\boldsymbol{\Sigma}}^{\text{adv}} \end{bmatrix}_{\ell,d} - \begin{bmatrix} I & \alpha I \\ \alpha I & I \end{bmatrix}_{\ell,d} \right)^2. \tag{8}$$

The total training objective of the model takes the following form,

$$\mathcal{L}(\mathbf{x}_{1,\ldots,N}, \mathbf{x}_{1,\ldots,N}^{\text{adv}}) = \lambda_{\text{REC}} \mathcal{L}_{\text{REC}}(\mathbf{x}_{1,\ldots,N}^{\epsilon}) + \lambda_{\text{KS}} \mathcal{L}_{\text{KS},k}^{\text{adv}}(\mathbf{z}_{1,\ldots,N}, \mathbf{z}_{1,\ldots,N}^{\text{adv}}) +$$
$$\lambda_{\text{CV}} \mathcal{L}_{\text{CV},k}^{\text{adv}}(\mathbf{z}_{1,\ldots,N}, \mathbf{z}_{1,\ldots,N}^{\text{adv}}). \tag{9}$$

The weights $\lambda_{\text{KS}}$ and $\lambda_{\text{CV}}$ can be calculated by taking the statistics of samples from the Gaussian prior [13], similar to eq. (10).

Table 1: Additional ablation study on MNIST images. Here $x_r$ refers to reference image, $x_a$ to adversarial image and $\tilde{x}_r$, $\tilde{x}_a$ to their corresponding reconstructions. The maximum input noise perturbation level $\lambda$ is limited to $1, 3$ and $5$. The best values observed for the ablation study on unimodal prior assumption are underlined. And the best values observed for the ablation on adversarial training method used are marked in bold.

| Method | Latent space attack | | | Maximum damage attack | | | FID($\downarrow$) |
|---|---|---|---|---|---|---|---|
| | MSSSIM($x_r, \tilde{x}_a$)($\downarrow$) | | | MSSSIM($\tilde{x}_r, \tilde{x}_a$)($\uparrow$) | | | |
| | 1 | 3 | 5 | 1 | 3 | 5 | |
| VAE | 0.55 | 0.78 | 0.89 | 0.64 | 0.27 | 0.08 | 43.21 |
| $\beta$-VAE | 0.52 | 0.73 | 0.86 | 0.65 | 0.36 | 0.22 | 42.72 |
| $\beta$-TCVAE | 0.53 | 0.69 | 0.83 | 0.73 | 0.38 | 0.28 | 45.61 |
| LipschitzVAE | 0.50 | 0.68 | 0.79 | 0.75 | 0.41 | 0.33 | 59.45 |
| SE | 0.49 | 0.62 | 0.68 | 0.90 | 0.60 | 0.54 | 47.34 |
| AVAE | 0.49 | 0.59 | 0.62 | 0.80 | 0.65 | 0.59 | 48.47 |
| Unimodal-Ours | 0.48 | 0.62 | 0.70 | 0.89 | 0.73 | 0.62 | 41.41 |
| PGD-Ours | 0.39 | **0.46** | **0.58** | **0.93** | **0.83** | **0.71** | **39.10** |
| Ours | **0.38** | 0.47 | 0.60 | 0.92 | 0.82 | 0.69 | 39.37 |

## A.2 Additional experimental results

**Additional robustness evaluation** In this section, we consider an additional metric to quantitatively evaluate the robustness of the model. To be precise, we compute the $l2$ distance between the reference image $\mathbf{x}_r$ and the reconstructions $\tilde{\mathbf{x}}_a$ of the adversarially perturbed variants $\mathbf{x}_a$. Similar to the main paper, under latent space attacks, we consider the target image $\mathbf{x}_r = \mathbf{x}_t$ as the reference images and report the $l2$ distance with the reconstruction $\tilde{\mathbf{x}}_a = f(\mathbf{z}_{\mathbf{x}+\delta})$ of the perturbed latent representation. For a maximum damage attack, we consider the original images $\mathbf{x}_r = \mathbf{x}$ as the reference images and consider the $l2$ distance with the reconstructions, $\tilde{\mathbf{x}}_a$. The observed values are given in Figure 1.

**Additional ablation study** We conduct two ablation studies in this section. First, we consider an expensive PGD-based adversarial learning in our training pipeline and report the evaluation metrics for MNIST images in Table 1. We observe a slight increase in the performance of our model when PGD adversarial learning is employed. However, it should be also noted that with PGD-based training, the computational time is two times more expensive than our original method. For the second experiment, we consider a unimodal prior assumption, to train our model and report the values for MNIST images in Table 1. It can be inferred that the unimodal version of the proposed method also performs better or is comparable with the other robust baseline models.

## A.3 Visual analysis of attacks

For qualitative analysis, we provide visual results for MNIST, FASHIONMNIST, SVHN, and CELEBA images for both adversarial attacks - latent space and maximum damage attacks. For latent space attacks, we compare the source image, clean reconstruction, adversarial image, adversarial reconstruction and target images across dataset as shown in Figures 2, 3, 4 and 5. For a successful attack, the adversarial reconstruction would strongly resemble the target image. As observed from the Figure, it can be seen that the proposed method remains more robust under latent space attacks when compared to the baseline models. For maximum damage attacks, we compare the source images, the corresponding clean reconstructions, adversarial images and their corresponding reconstructions across dataset as shown in Figures 6, 7, 8 and 9. These attacks are more successful when the adversarial reconstructions are less similar in appearance to the clean reconstructions. As shown in the Figure, we observe that both of these attacks get more successful with an increase in noise perturbation. However, for a given noise perturbation the proposed method is more robust when compared to other models.

### A.4 Experimental details

**Network architectures**  We use a consistent network architecture for the encoder-decoder pair during training. For MNIST and FASHIONMNIST images, we train a fully connected network architecture, a 4-layer multi-layer perceptron (MLP) with 200 neurons and ReLU activation at each layer. For SVHN and CELEB images we use a convolution network architecture similar to [13]. The encoder includes a 4-layer convolution network with the number of output channels $(128, 256, 512, 1024)$ respectively, with strides equal to 2 and a kernel size of $(4, 4)$. And the decoder comprises a 4-layer de-convolution network with the number of output channels $(1024, 512, 256, 128)$ respectively, with strides equal to 2 and a kernel size of $(4, 4)$. The latent space dimension of 10 is used for MNIST and FASHIONMNIST images, 100 for SVHN, and 64 for CELEBA images.

**Dataset**  For empirical analysis, we consider four image benchmark datasets: MNIST (GNU General Public License v3.0) [9], FASHIONMNIST (MIT License) [16], SVHN (GNU General Public License v3.0) [12] and CELEBA (non-commercial research) [10]. Similar to [13], for MNIST, FASHIONMNIST, and SVHN, images of size $32 \times 32$ are used. For CELEBA, images resized to $64 \times 64$ after center cropping to $140 \times 140$ are used.

**Implementation details**  We carried out all the experiments on a single GTX1080 GPU with 16 GB RAM. All the conducted experiments were part of a carbon-neutral framework-based GPU cluster and hence did not contribute to climate change. We will open-source the code to implement the proposed method upon acceptance of the paper.

For training the encoder-decoder network of our model, we utilize an ADAM [7] optimizer with a batch size of 100 and an initial learning rate of 0.002 with exponential decay based on the validation loss. For the multi-modal prior definition, we follow the same setup as in [13]. The coupling parameter is chosen as 0.95 for MNIST and FASHIONMNIST images, and 0.92 for SVHN and CELEBA images. For the classification downstream application, we train a simple two-layer MLP-based classifier. The network is trained for 25 epochs with an ADAM [7] based optimizer with a learning rate of 0.01, batch size of 100.

To avoid expensive hyperparameter tuning of the loss hyperparameters $\lambda_{\text{KS}}$ and $\lambda_{\text{CV}}$ mentioned in the training objective of the model in eq:11(main paper), we extend the simple heuristic mentioned in [13] to our model definition. Consider a target GMM prior, for a model trained with latent space dimension $D$ and batch size $N$, the loss hyperparameters are calculated as follows, for $m = 1, \ldots, M$ samples $\mathbf{z}_1^{(m)}, \ldots, \mathbf{z}_N^{(m)}$ and $\mathbf{z}'^{(m)}_1, \ldots, \mathbf{z}'^{(m)}_N$ independently sampled from the prior GMM,

$$\lambda_{\text{KS}}^{-1} = \frac{1}{M} \mathcal{L}_{\text{KS}} \left( \mathbf{z}_{1,\ldots,N}^{(m)}, \mathbf{z}'^{(m)}_{1,\ldots,N} \right), \qquad \lambda_{\text{CV}}^{-1} = \frac{1}{M} \mathcal{L}_{\text{CV}} \left( \mathbf{z}_{1,\ldots,N}^{(m)}, \mathbf{z}'^{(m)}_{1,\ldots,N} \right). \tag{10}$$

In our experiments, we consider a simple approach by independently sampling from the GMM prior instead of the coupled prior definition since we observed good empirical results with this approximation. The hyperparameter $\lambda_{\text{REC}}$ can be approximated as the inverse of the best obtained loss obtained by training an autoencoder model.

For a fair comparison, we use similar architecture for all the baseline models considered. We used the Pytorch implementation in the Githib repository[1] for training the baseline models, VAE [8], $\beta-$VAE [6] and $\beta-$TCVAE [4] . For LipschitzVAE [1] we used the official Pytorch implementation[2]. For SE [3], we re-implemented the method in Pytorch and for AVAE [2] we reimplemented a Pytorch version of the official JAX-based version[3]. Since the official Github implementation for AVAEs only provides an MLP-based training pipeline, we only report AVAE results for MNIST and FASHIONMNIST images.

**Evaluation setup**  To evaluate the robustness of the model, we consider mainly two types of adversarial attacks, latent space attacks, and maximum damage attacks. For experimental analysis, under these attacks, we consider 100 randomly chosen test images from the corresponding dataset and run 10 simulations to report the results. The noise perturbation levels of $1, 3$ and $5$ are chosen. While choosing the target image for latent space attacks, we explicitly choose an image from a different

---

[1]`https://github.com/YannDubs/disentangling-vae`
[2]`https://github.com/FabianBarrett/Lipschitz_VAEs`
[3]`https://github.com/deepmind/deepmind-research/tree/master/avae`

Table 2: Sensitivity analysis of the number of modes in the GMM prior on MNIST images.

| Number of modes | Latent space attack | | | | | | Maximum damage attack | | | | | | FID($\downarrow$) |
|---|---|---|---|---|---|---|---|---|---|---|---|---|---|
| | MSSSIM($\mathbf{x}_r, \tilde{\mathbf{x}}_a$)($\downarrow$) | | | MSSSIM($\mathbf{x}_r, \mathbf{x}_a$)($\downarrow$) | | | MSSSIM($\tilde{\mathbf{x}}_r, \tilde{\mathbf{x}}_a$)($\uparrow$) | | | MSSSIM($\mathbf{x}_r, \mathbf{x}_a$)($\downarrow$) | | | |
| | 1 | 3 | 5 | 1 | 3 | 5 | 1 | 3 | 5 | 1 | 3 | 5 | |
| 1 | 0.48 | 0.62 | 0.70 | 0.99 | 0.93 | 0.86 | 0.89 | 0.73 | 0.62 | 0.98 | 0.92 | 0.86 | 42.45 |
| 5 | 0.43 | 0.53 | 0.66 | 0.98 | 0.92 | 0.83 | 0.91 | 0.76 | 0.65 | 0.98 | 0.91 | 0.82 | 40.11 |
| 10 | 0.38 | 0.47 | 0.60 | 0.95 | 0.90 | 0.80 | 0.92 | 0.82 | 0.69 | 0.98 | 0.89 | 0.78 | 39.37 |
| 15 | 0.37 | 0.45 | 0.58 | 0.95 | 0.89 | 0.80 | 0.93 | 0.82 | 0.70 | **0.97** | 0.89 | 0.77 | 39.04 |
| 20 | 0.37 | 0.45 | **0.57** | **0.94** | **0.88** | 0.79 | 0.93 | **0.83** | **0.71** | 0.97 | **0.88** | 0.78 | 38.49 |
| 25 | **0.36** | **0.43** | 0.58 | **0.94** | **0.88** | 0.78 | 0.94 | **0.83** | **0.71** | 0.97 | **0.88** | 0.77 | **38.02** |

Table 3: Sensitivity analysis of the hyperparameter alpha on MNIST images.

| Coupling parameter $\alpha$ | Latent space attack | | | | | | Maximum damage attack | | | | | | FID($\downarrow$) |
|---|---|---|---|---|---|---|---|---|---|---|---|---|---|
| | MSSSIM($\mathbf{x}_r, \tilde{\mathbf{x}}_a$)($\downarrow$) | | | MSSSIM($\mathbf{x}_r, \mathbf{x}_a$)($\downarrow$) | | | MSSSIM($\tilde{\mathbf{x}}_r, \tilde{\mathbf{x}}_a$)($\uparrow$) | | | MSSSIM($\mathbf{x}_r, \mathbf{x}_a$)($\downarrow$) | | | |
| | 1 | 3 | 5 | 1 | 3 | 5 | 1 | 3 | 5 | 1 | 3 | 5 | |
| 0.1 | 0.46 | 0.56 | 0.68 | 0.98 | 0.91 | 0.84 | 0.80 | 0.59 | 0.50 | 0.98 | 0.91 | 0.80 | 40.18 |
| 0.3 | 0.44 | 0.55 | 0.65 | 0.97 | 0.92 | 0.82 | 0.81 | 0.65 | 0.61 | 0.97 | 0.90 | 0.80 | 39.84 |
| 0.5 | 0.43 | 0.53 | 0.64 | 0.97 | 0.90 | 0.82 | 0.85 | 0.75 | 0.63 | 0.98 | 0.89 | 0.80 | 40.01 |
| 0.7 | 0.40 | 0.49 | 0.63 | 0.96 | 0.90 | 0.81 | 0.90 | 0.79 | 0.67 | 0.98 | 0.89 | 0.79 | 39.28 |
| 0.9 | 0.39 | 0.48 | 0.62 | 0.95 | 0.90 | 0.80 | 0.91 | 0.81 | 0.68 | 0.98 | 0.89 | 0.78 | 39.61 |
| 0.95 | **0.38** | **0.47** | 0.60 | **0.95** | 0.90 | **0.80** | **0.92** | 0.82 | **0.69** | 0.98 | **0.89** | 0.78 | **39.37** |
| 1.0 | **0.38** | **0.47** | 0.59 | **0.95** | 0.89 | **0.80** | **0.92** | 0.83 | **0.69** | 0.97 | 0.87 | 0.78 | 41.86 |

class than of original image for MNIST, FASHIONMNIST, and SVHN images. For evaluating the fidelity of the learned representations, we report Fréchet Inception Distance (FID) [5] of the generated samples in the main paper. We calculate the FID between 10000 generated images and validation images for the corresponding dataset and report the average value obtained after five different runs. The FIDs observed for the proposed method along with error bars (for different runs) are as follows, MNIST: $39.37 \pm 0.9$, FASHION-MNIST: $64.89 \pm 0.9$, SVHN: $38.89 \pm 1.2$ and CELEBA: $51.98 \pm 1.3$. We report the error bar for all the robustness evaluation metrics reported in Table 1 in the main paper in Figure 10. The observed attack loss similar to Figure.2 in the main paper for MNIST images is shown in Figure 11.

## A.5 Sensitivity analysis of the number of modes in the GMM prior

The number of modes in the chosen prior is a hyperparameter of the proposed model. Hence we report a sensitivity analysis of the number of modes of the GMM prior and the observed robustness of the model. We analyze the robustness and generation performance of our model on MNIST images for different number of components in the chosen prior in Table 2. We use the same number of modes used in Saseendran et.al for all our experiments. As expected from previous analysis in Saseendran et.al., with an increased number of modes in the GMM prior the generation performance of our extended model also improved. Most importantly, we observe a similar trend for robustness as well. That is, with a higher number of components in the chosen GMM prior the model exhibits improved robustness.

## A.6 Sensitivity analysis of the Coupling strength

In this section, we study the sensitivity of our model towards the coupling strength $\alpha$ (see Table 3). We observe that a larger coupling strength $\alpha$ leads to improved robustness against both latent space and maximum damage attacks. However, as mentioned in the limitations section a strong coupling strength, i.e. $\alpha = 1$, compromises the generation fidelity of the model. In our experiments, we tuned the coupling strength on each individual dataset. We observed that a coupling strength in the range of $0.9 \leq \alpha < 1$ yields the best trade-off between generation and robustness across all datasets. In our experiments, we chose $\alpha = 0.95$ for MNIST and FASHIONMNIST images, and $\alpha = 0.92$ for SVHN and CELEBA images.

## A.7 Additional evaluation of the decoder quality

In this section, we further study the quality of the decoder of the proposed model. Here we evaluate the MSSSIM between the reference images $x_r$ and its reconstructions $\tilde{x}_r$ and the adversarial images $x_a$ and its corresponding reconstructions $\tilde{x}_a$ for both attack modes. The results are reported in the Table 4. Compared to the non-robust variants (VAE, $\beta$-VAE, $\beta$-TCVAE), the quality of the reconstructions of the reference images is compromised in robust VAE models (LipschitzVAE, SE, AVAE), whereas the proposed model exhibits comparatively better performance. This further aligns with the observation that our model yields better reconstruction fidelity than all the baselines. Further, we observe higher similarity between the adversarial images and their reconstructions for all robust VAE models. This is due to the fact that all these models employ adversarial training.

Table 4: Decoder quality - Similarity between images and its corresponding reconstructions. We consider the MSSSIM between the reference image($\mathbf{x}_r$) and its reconstruction($\tilde{\mathbf{x}}_r$) and the adversarial image($\mathbf{x}_a$) and its reconstruction($\tilde{\mathbf{x}}_a$) for both latent space and maximum damage attack. For latent space attack the reference image is the target image and for the maximum damage attack the reference image is the input image.

| MNIST | MSSSIM($\mathbf{x}_r, \tilde{\mathbf{x}}_r$)($\uparrow$) | Latent space attack MSSSIM($\mathbf{x}_a, \tilde{\mathbf{x}}_a$)($\uparrow$) | | | Maximum damage attack MSSSIM($\mathbf{x}_a, \tilde{\mathbf{x}}_a$)($\uparrow$) | | |
|---|---|---|---|---|---|---|---|
| | | 1 | 3 | 5 | 1 | 3 | 5 |
| VAE | 0.94 | 0.71 | 0.42 | 0.24 | 0.66 | 0.35 | 0.31 |
| $\beta$-VAE | 0.93 | 0.73 | 0.48 | 0.31 | 0.64 | 0.38 | 0.35 |
| $\beta$-TCVAE | 0.93 | 0.72 | 0.47 | 0.25 | 0.64 | 0.40 | 0.38 |
| LipschitzVAE | 0.85 | 0.70 | 0.46 | 0.38 | 0.66 | 0.44 | 0.39 |
| SE | 0.91 | 0.71 | 0.53 | 0.48 | 0.69 | 0.58 | 0.50 |
| AVAE | 0.92 | 0.70 | 0.55 | 0.50 | 0.71 | 0.62 | 0.59 |
| Ours | **0.97** | **0.89** | **0.79** | **0.55** | **0.82** | **0.70** | **0.61** |

| FASHIONMNIST | MSSSIM($\mathbf{x}_r, \tilde{\mathbf{x}}_r$)($\uparrow$) | Latent space attack MSSSIM($\mathbf{x}_a, \tilde{\mathbf{x}}_a$)($\uparrow$) | | | Maximum damage attack MSSSIM($\mathbf{x}_a, \tilde{\mathbf{x}}_a$)($\uparrow$) | | |
|---|---|---|---|---|---|---|---|
| | | 1 | 3 | 5 | 1 | 3 | 5 |
| VAE | 0.88 | 0.61 | 0.52 | 0.50 | 0.45 | 0.33 | 0.29 |
| $\beta$-VAE | 0.87 | 0.61 | 0.57 | 0.51 | 0.53 | 0.26 | 0.29 |
| $\beta$-TCVAE | 0.88 | 0.62 | 0.59 | 0.53 | 0.54 | 0.23 | 0.31 |
| LipschitzVAE | 0.84 | 0.77 | 0.58 | 0.55 | 0.58 | 0.29 | 0.35 |
| SE | 0.86 | 0.78 | 0.67 | 0.60 | 0.67 | 0.65 | 0.48 |
| AVAE | 0.87 | **0.80** | 0.75 | 0.61 | 0.85 | 0.65 | 0.47 |
| Ours | **0.91** | 0.79 | **0.76** | **0.62** | **0.89** | **0.71** | **0.56** |

| SVHN | MSSSIM($\mathbf{x}_r, \tilde{\mathbf{x}}_r$)($\uparrow$) | Latent space attack MSSSIM($\mathbf{x}_a, \tilde{\mathbf{x}}_a$)($\uparrow$) | | | Maximum damage attack MSSSIM($\mathbf{x}_a, \tilde{\mathbf{x}}_a$)($\uparrow$) | | |
|---|---|---|---|---|---|---|---|
| | | 1 | 3 | 5 | 1 | 3 | 5 |
| VAE | 0.85 | 0.71 | 0.62 | 0.57 | 0.68 | 0.58 | 0.50 |
| $\beta$-VAE | 0.84 | 0.72 | 0.60 | 0.58 | 0.68 | 0.57 | 0.52 |
| $\beta$-TCVAE | 0.85 | 0.71 | 0.61 | 0.59 | 0.69 | 0.58 | 0.51 |
| LipschitzVAE | 0.80 | 0.74 | 0.63 | 0.58 | 0.68 | 0.60 | 0.54 |
| SE | 0.83 | 0.79 | 0.69 | 0.62 | 0.82 | 0.78 | 0.62 |
| Ours | **0.90** | **0.81** | **0.73** | **0.66** | **0.84** | **0.80** | **0.65** |

| CELEBA | MSSSIM($\mathbf{x}_r, \tilde{\mathbf{x}}_r$)($\uparrow$) | Latent space attack MSSSIM($\mathbf{x}_a, \tilde{\mathbf{x}}_a$)($\uparrow$) | | | Maximum damage attack MSSSIM($\mathbf{x}_a, \tilde{\mathbf{x}}_a$)($\uparrow$) | | |
|---|---|---|---|---|---|---|---|
| | | 1 | 3 | 5 | 1 | 3 | 5 |
| VAE | 0.84 | 0.79 | 0.72 | 0.65 | 0.76 | 0.71 | 0.65 |
| $\beta$-VAE | 0.83 | 0.74 | 0.71 | 0.67 | 0.75 | 0.73 | 0.69 |
| $\beta$-TCVAE | 0.83 | 0.73 | 0.68 | 0.64 | 0.75 | 0.70 | 0.68 |
| LipschitzVAE | 0.79 | 0.74 | 0.70 | 0.67 | 0.75 | 0.72 | 0.69 |
| SE | 0.80 | 0.79 | 0.75 | **0.70** | 0.78 | 0.76 | 0.74 |
| Ours | **0.86** | **0.80** | **0.77** | **0.70** | **0.80** | **0.79** | **0.77** |

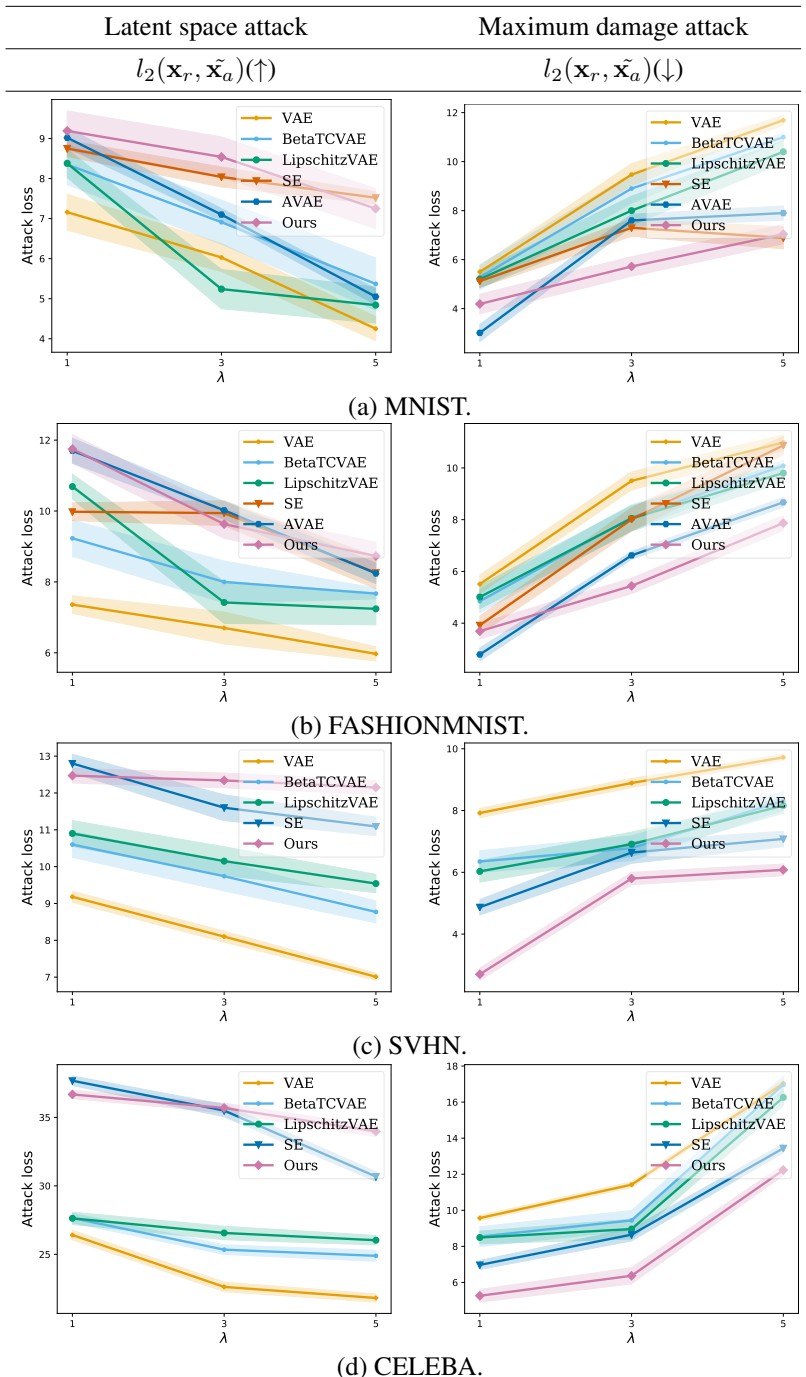

Figure 1: Observed *l*2 distance between images in the event of latent space and maximum damage attacks across datasets. Here, randomly chosen $100$ test images are attacked in $10$ different trials. $\mathbf{x}_r$ refers to reference image and $\tilde{\mathbf{x}}_a$ to the corresponding reconstruction of the adversarial image $\mathbf{x}_a$. The maximum input noise perturbation levels $\lambda$ are limited to $1, 3$ and $5$.

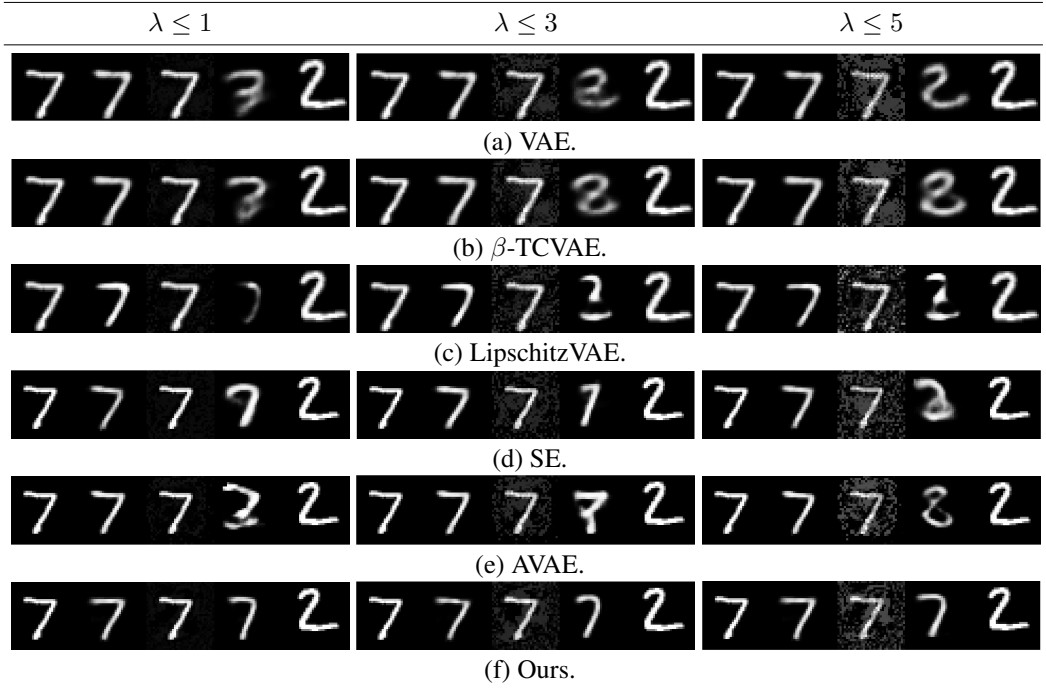

Figure 2: Visual appraisal of latent space attacks - Qualitative analysis across models on MNIST images with maximum input noise perturbation level $\lambda$ limited to $1, 3$ and $5$. (from left to right)Images in each row correspond to source image, clean reconstruction, adversarial image, adversarial reconstruction and target image respectively.

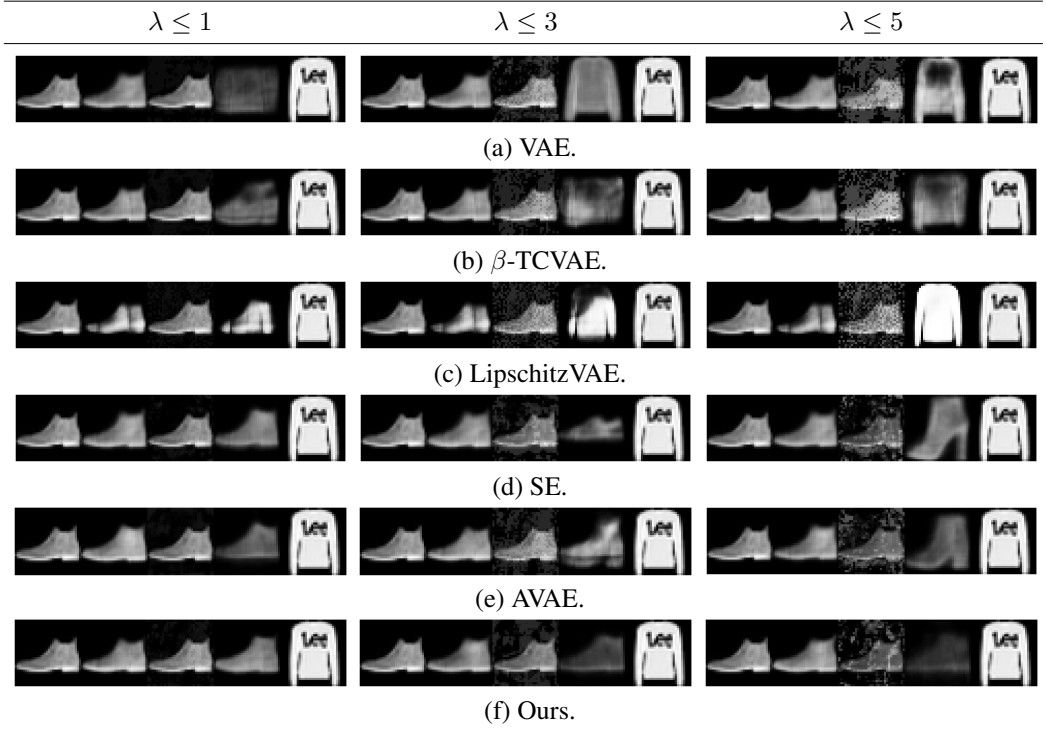

Figure 3: Visual appraisal of latent space attacks - Qualitative analysis across models on FASH-IONMNIST images with maximum input noise perturbation level $\lambda$ limited to $1, 3$ and $5$. (from left to right) Images in each row correspond to source image, clean reconstruction, adversarial image, adversarial reconstruction and target image respectively.

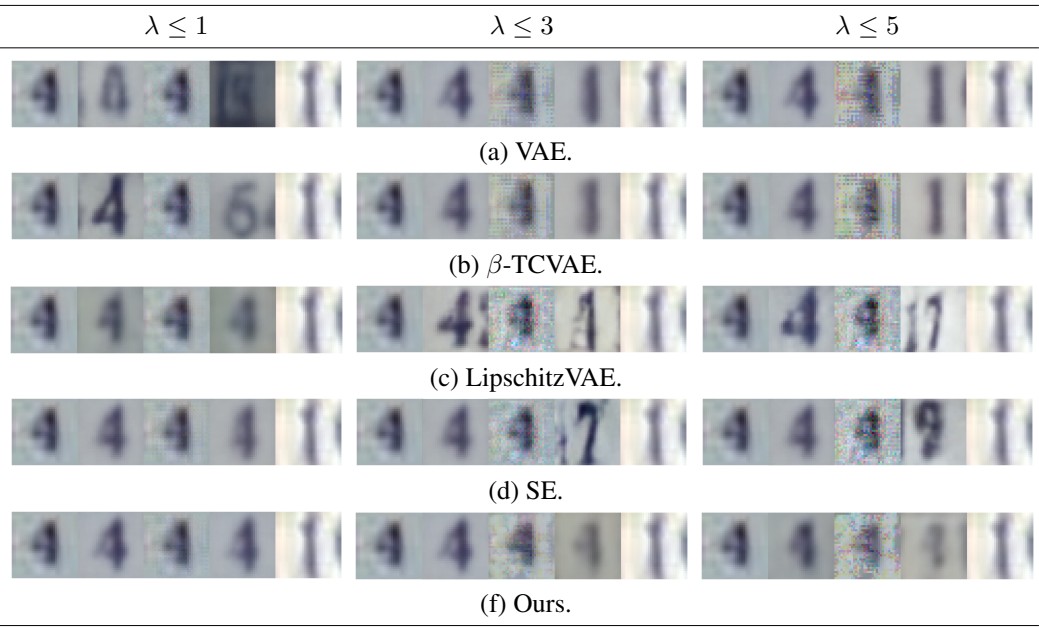

Figure 4: Visual appraisal of latent space attacks - Qualitative analysis across models on SVHN images with maximum input noise perturbation level $\lambda$ limited to $1, 3$ and $5$. (from left to right) Images in each row correspond to source image, clean reconstruction, adversarial image, adversarial reconstruction and target image respectively.

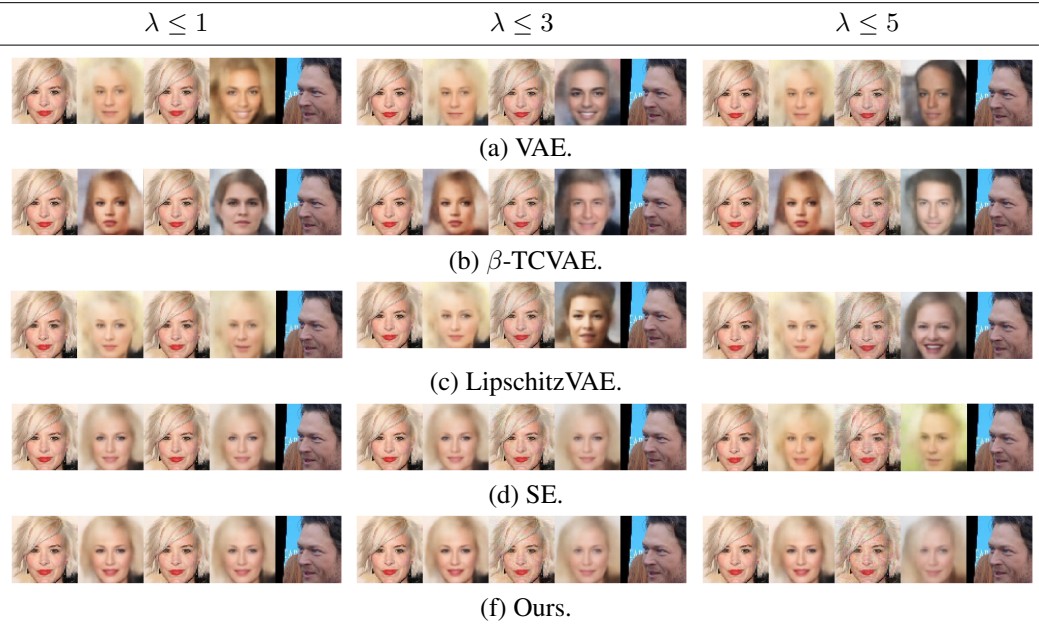

| $\lambda \leq 1$ | $\lambda \leq 3$ | $\lambda \leq 5$ |

(a) VAE.

(b) $\beta$-TCVAE.

(c) LipschitzVAE.

(d) SE.

(f) Ours.

Figure 5: Visual appraisal of latent space attacks - Qualitative analysis across models on CELEBA images with maximum input noise perturbation level $\lambda$ limited to $1, 3$ and $5$. (from left to right) Images in each row correspond to source image, clean reconstruction, adversarial image, adversarial reconstruction and target image respectively.

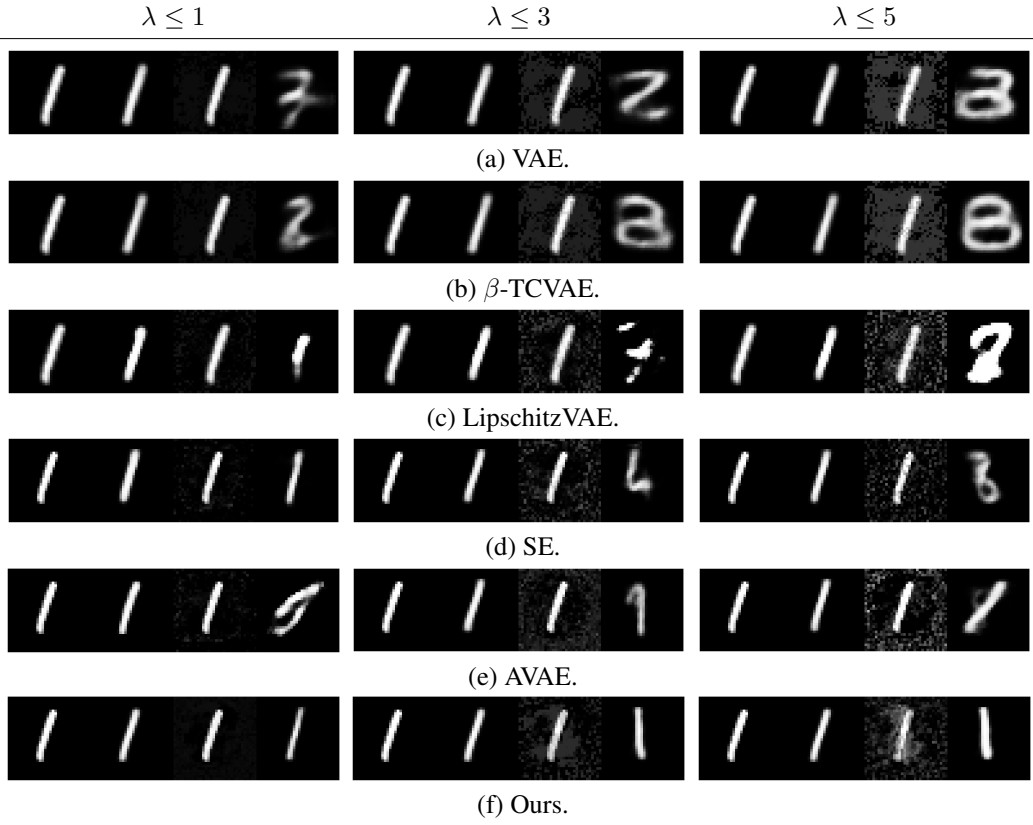

Figure 6: Visual appraisal of maximum damage attacks - Qualitative analysis across models on MNIST images with maximum input noise perturbation level $\lambda$ limited to $1, 3$ and $5$. (from left to right) Images in each row correspond to source image, clean reconstruction, adversarial image and adversarial reconstruction respectively.

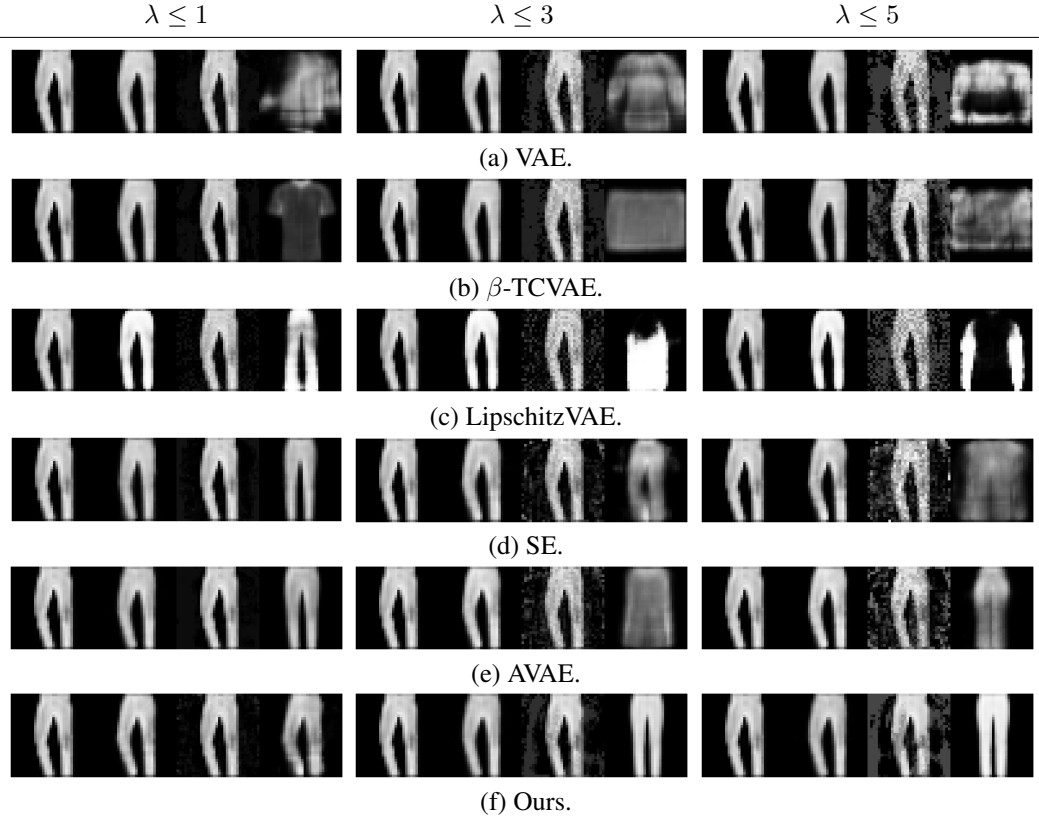

Figure 7: Visual appraisal of maximum damage attacks - Qualitative analysis across models on FASHIONMNIST images with maximum input noise perturbation level $\lambda$ limited to 1, 3 and 5. (from left to right) Images in each row correspond to source image, clean reconstruction, adversarial image and adversarial reconstruction respectively.

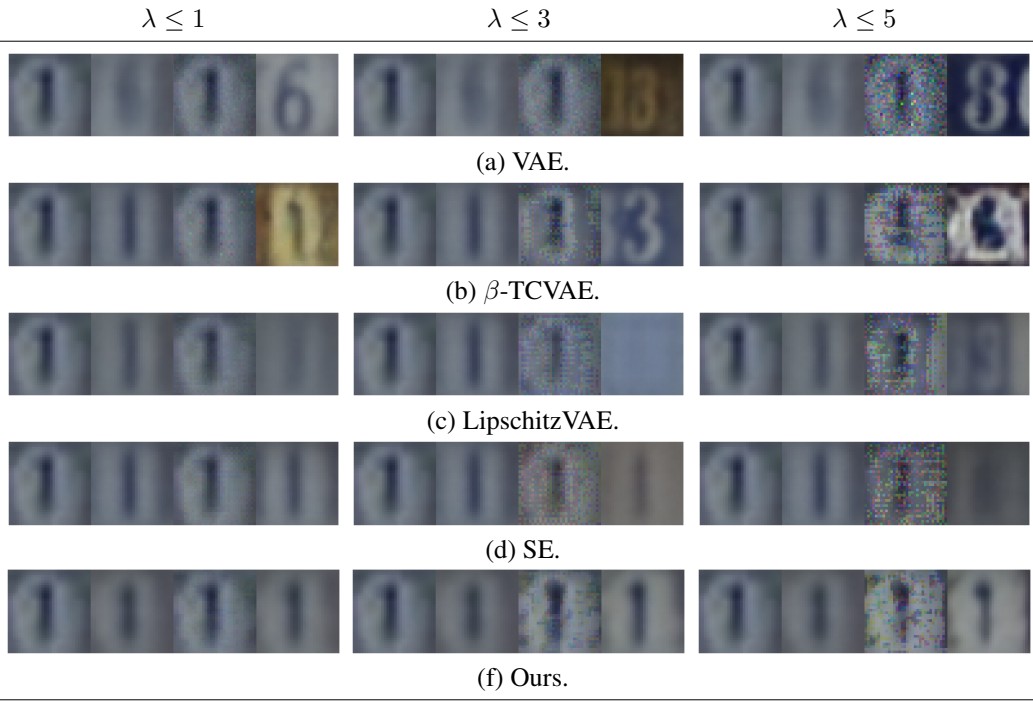

Figure 8: Visual appraisal of maximum damage attacks - Qualitative analysis across models on SVHN images with maximum input noise perturbation level $\lambda$ limited to $1, 3$ and $5$. (from left to right) Images in each row correspond to source image, clean reconstruction, adversarial image and adversarial reconstruction respectively.

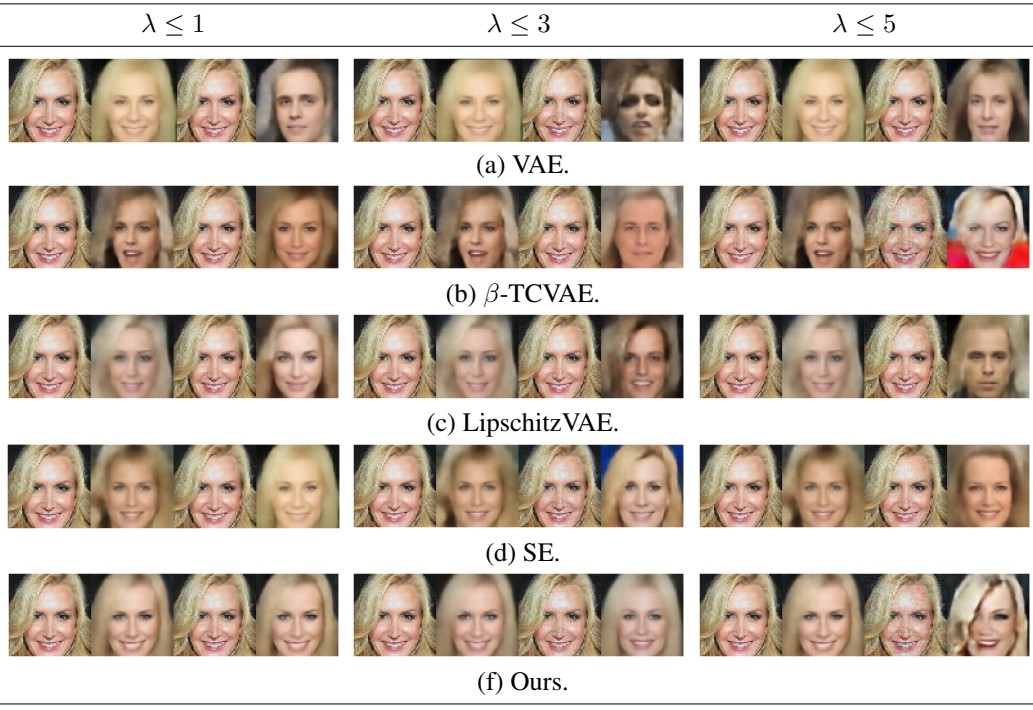

Figure 9: Visual appraisal of maximum damage attacks - Qualitative analysis across models on CELEBA images with maximum input noise perturbation level $\lambda$ limited to $1, 3$ and $5$. (from left to right) Images in each row correspond to source image, clean reconstruction, adversarial image and adversarial reconstruction respectively.

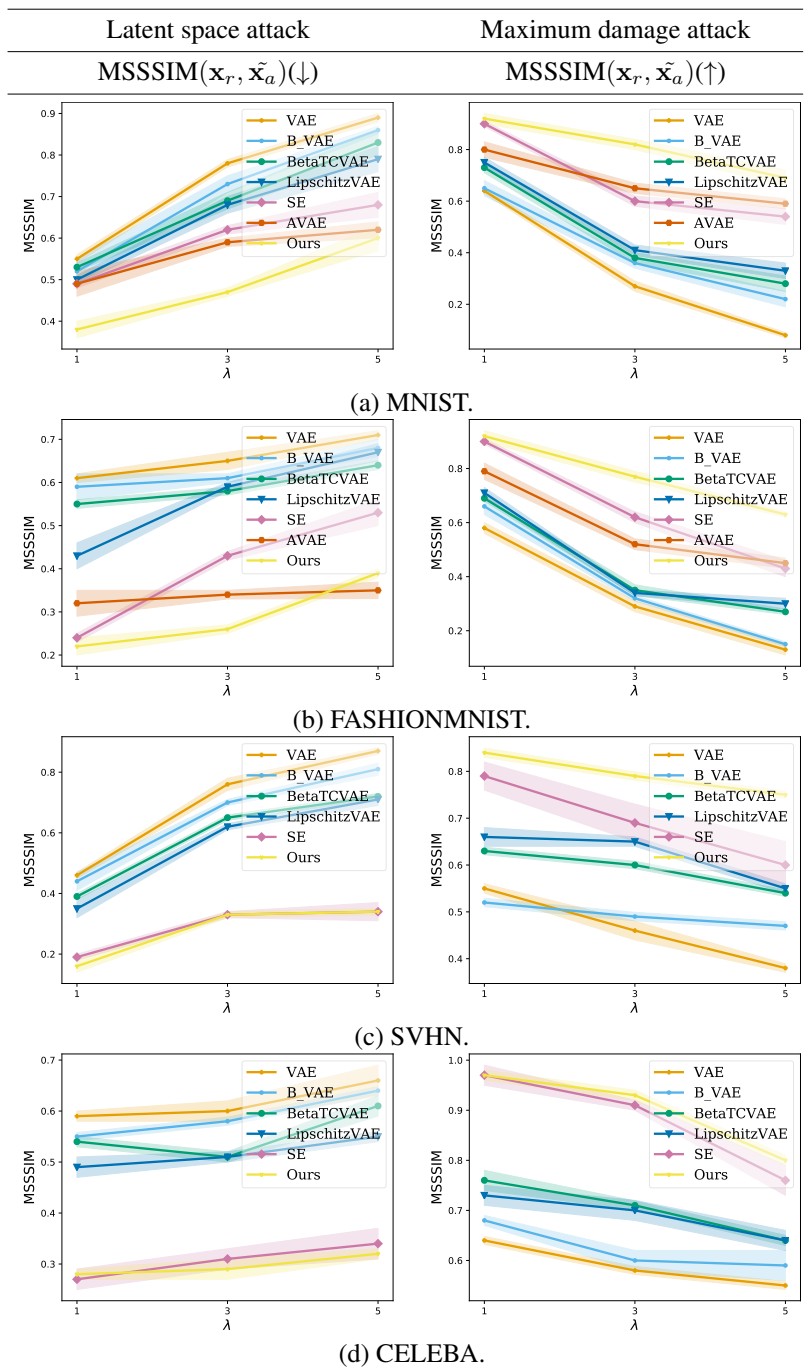

Figure 10: Observed MSSSIM with error bar between images in the event of latent space and maximum damage attacks across dataset. Here randomly chosen 100 test images are attacked in 10 different trials and we report the mean and standard deviation. $\mathbf{x}_r$ refers to reference image and $\tilde{\mathbf{x}_a}$ to the corresponding reconstruction of the adversarial image $\mathbf{x}_a$. The maximum input noise perturbation levels $\lambda$ are limited to $1, 3$ and $5$.

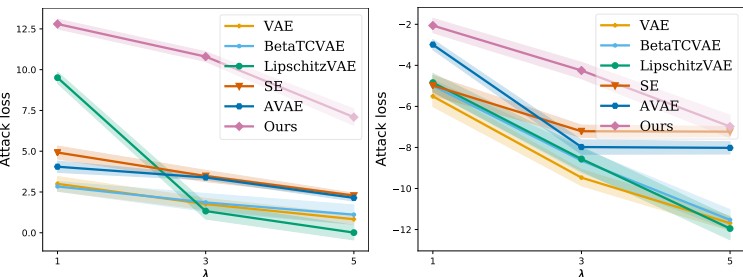

Figure 11: Observed attack losses for (a) latent space attack and (b) maximum damage attack with varying $\lambda$ values for MNIST images. We report the observed mean and standard deviation by attacking 100 randomly chosen test images in 10 different trials. Higher loss indicates more robustness.