# OpenReview forum: "Trading off Image Quality for Robustness is not Necessary with Regularized Deterministic Autoencoders"
_NeurIPS.cc/2022/Conference — NeurIPS 2022 Accept_

### Official Review · Reviewer_ayNS · 2022-07-11

**Rating:** 6
**Confidence:** 3
**Soundness:** 3 good
**Presentation:** 3 good
**Contribution:** 2 fair

**Summary:**

The paper proposes new approaches for improving the robustness of Variational Autoencoders (VAEs). The paper finds that deterministic autoencoder with multimodal prior (proposed in prior work) not only improves the fidelity of generated samples, but is also more robust to adversarial attacks. On top of that, the paper proposes to incorporate adversarial samples into the training process, and add regularizations to ensure that the latents from the adversarially perturbed samples also follow the same prior distribution. Results show that the proposed method achieves better fidelity and robustness across several datasets.

**Questions:**


I appreciate that the authors point out the downside of having a new hyperparameter alpha. However, I think it is still important to at least mention how alpha was tuned for all the experiments, and provide a sensitivity analysis of alpha. This would make it easier for people to adopt the method. I will adjust the score according to the response to this question.



My other questions are:

* Table 1: I don't understand how you get x_r and x_a for maximum damage attack, and why lower MSSSIM(x_r,x_a) is better. I could not find related explanations or descriptions in the main text.

* Line 158: z_n and z_n^{adv} are undefined. I suppose they are g(x_n) and g(x_n + \delta_{x_n})?

* Line 162: Does "ignoring cross-covariance between z_{1,...,n} and z_{1,...,n}^{adv}" mean the 0s in last matrix in Eq 8? Please make it more explicit

* Eq. 6: g is undefined. I suppose it means the encoder?



Some other minor typos and questions:

* The title of the citation [26] is missing.

* Figure 1 and its caption are hard to understand by themselves. It becomes clearer after I read Section 3.2, which appears much later in this paper.

* Should the denominators of Eq 8 and Eq 10 be 4D^2 instead of 2D^2?

* Line 171: left -> right

* Table 2: Consider putting the best metrics of each column in bold?




**Limitations:**


The paper adequately addressed the limitations. The paper did not discuss the potential negative societal impact of the work.

**Strengths And Weaknesses:**


Originality: The key techniques of this paper (incorporating adversarial samples in training, KS test) are simple adaptations from prior work. However, this specific combination for improving the robustness of VAE is interesting.

Quality: Overall, the paper is technically sound, and most of the claims are well-supported by experiments.

Clarity: The paper is easy to read. However, there are some typos and some unclear statements. See questions below.

Significance: The paper could be of interest to practitioners as it shows that the proposed technique could improve the robustness without sacrificing fidelity.

---

> ### Author Response · Authors · 2022-08-02
> **Sensitivity analysis of coupling strength**
>
> 1. Sensitivity analysis of the hyperparameter $\alpha$ - Thank you for suggesting this experiment.
> We have summarized the results of the sensitivity of our model towards the coupling strength $\alpha$ in the Table 3, section A.6 in the Appendix.
> We observe that a larger coupling strength $\alpha$ leads to improved robustness against both latent space and maximum damage attacks.
> However, as mentioned in the limitations section a strong coupling strength, i.e. $\alpha=1$, compromises the generation fidelity of the model.
> In our experiments, we tuned the coupling strength of each individual dataset.
> We observed that a coupling strength in the range of $0.9\leq\alpha<1$ yields the best trade-off between generation and robustness across all datasets.
> In our experiments, we chose $\alpha=0.95$ for MNIST and FASHIONMNIST images, and $\alpha=0.92$ for SVHN and
> CELEBA images.
>
> 2. Maximum damage attack - The adversarial image $x_a$ is the image obtained by adding an $\epsilon$-perturbation to the original input image $x=x_r$ in the case of maximum damage attacks.
> We assume that the amount of noise/local change needed to cause maximum damage scales with the model robustness. Specifically, to attack a robust model, an attacker would need to impose more change on the input $x$. Therefore, MSSSIM$(x_a,x_r)$ should be low for robust models.
> We will elaborate on the explanation in the main paper to clarify this point.
>
> 3.  Line 158 - $z_n$ and $z_n^{adv}$ are the latent representation of the input ($x_n$) and its corresponding adversarial sample ($x_n + \delta_{x_n}$).
> We will add this definition to the main paper accordingly.
>
> 4. line 162 -
> As you pointed out correctly, when "ignoring the cross-covariance between the latents $z_{1,...,n}$ and $z_{1,...,n}^{adv}$", the off-diagonal elements in the covariance matrix of the GMM prior, that is the last matrix mentioned in equation (8), are zero. We will clarify this point in the revised version of the paper.
>
> 5. Eqn 6 - Indeed, $g$ denotes the encoder of the model! We will include this definition in the corresponding section.
>
> 6. Minor questions - We have included the missing title of the mentioned reference and indicated the best metrics in Table 2 in bold in the revised version.
>
> 7. Figure 1 explanation -  We have edited Figure~1 and its title in the revised version of the paper.
> Most importantly, in the new version, we visually distinguish between the adversarial examples before and after regularization to visualize the effect of joint regularization of the original and adversarial samples.
>
> 8. Denominators of Eq 8 and Eq 10 - Yes, that is absolutely correct. Thanks for pointing this out! We have corrected this error in the revised version of the paper.

---

> > ### Comment · Reviewer_ayNS · 2022-08-07
> > **Comments**
> >
> > Thank the authors for the detailed explanations! The response addresses all my concerns.
> >
> > The added experiments on the sensitivity analysis of alpha on MNIST are very helpful for readers who want to try your methods. My only suggestion is that, since in the updated Appendix it is suggested that hyper-parameter tuning for alpha was done for each dataset and the optimal alphas are different across datasets, it would be even better and more informative to put the results of table 3 for all datasets. I would assume it does not require too much work as you already have the data. Nevertheless, the current results are sufficient, and it is fine not to add them.
> >
> > Thank you for the clarifications/confirmation on other questions as well.
> >
> > Because of these, I will increase the score.

---

### Official Review · Reviewer_Xb52 · 2022-07-11

**Rating:** 7
**Confidence:** 2
**Soundness:** 3 good
**Presentation:** 3 good
**Contribution:** 3 good

**Summary:**

The authors extend existing methods for training variational autoencoders to be robust against adversarial attacks. They implement this by training on adversarial examples during training. They show that the resulting networks are less susceptible to adversarial attacks have better MSSSIM metrics and finally that the latent representations are better suited for downstream classification tasks.

**Questions:**

The authors claim that their method does not increase the computational complexity, yet they use adversarial examples during training. Should the generation of the adversarial examples not increase the computational complexity?

The authors should provide more details on the sections "ablation study" and "robustness to downstream applications". Here the text is not clear what they did. I do not understand what the "ablation study" is exactly about and for the downstream applications it might help to know what kind of MLP the authors are using.

The authors present how their method improves the robustness against adversarial attacks but they show little how this affects the baseline performance. How is the MSSSIM for x_r and x_r tilde? Does the new approach provide similar decoded image quality? Also would not MSSSIM for x_a and x_a tile be interesting, how well the adversarial images can be reconstructed by the autoencoder?

The authors say that the adversarial samples should not explore unexplored regions of the latent space. I do not quite understand why this is beneficial. Shouldn't the newly generated samples try to explore more regions then the normal samples already explore?

**Limitations:**

The authors discuss the limitations of their work mainly in terms of computational power and additional hyper parameters that need to be tuned and how datasets with different distributions might not work as well with the approach.

The authors do not discuss potential negative societal impact of their work and just state N/A. Surely if their work is relevant, it should have a potential societal impact of which a part can also be negative. The authors could elaborate a bit on this to address this point.

**Strengths And Weaknesses:**

Here I have to admit that I am not very familiar with the current literature and the math behind the statistics of variational autoencoders. I have some difficulties understanding the exact novelty of the approach the authors have taken. It seems the basic idea is to prevent adversarial input examples during training and ensure some statistical properties of these examples.
But I assume the method is explained well for a reader with the sufficient background.

What comes a bit short tough is the explanation of the experiments under the headers "Ablation Study" and "Robustness to downstream applications". There the authors seem to omit the details.

The results seem to lead to a significant improvement in the results.

---

> ### Author Response · Authors · 2022-08-02
> **Additional evaluation of decoder quality and clarification on experiments**
>
>
> 1. Ablation study  - In the ablation study in Section 4 of the main paper, we compare three different variants of regularized deterministic autoencoders to evaluate the importance of joint regularization of the original and adversarial samples.
> We begin with the model proposed by Saseendran et.al., which we denote as GMM-DAE.
> Second, we study the *augmented* model defined by equations (7) and (8) in Section 3.2 of the main paper, but *without* the coupling of original and adversarial latent representations (Augmented).
> We compare the robustness and fidelity of these models with our proposed model, i.e. the final variant, where original and adversarial latent representations are not only regularized towards the same prior but coupled according to equations (9), (10), and (11) in Section 3.3 (Ours).
> The observed metrics are reported in Table 2 in the main paper.
> We observe that the proposed method yields comparatively better performance in terms of robustness while still maintaining the generation fidelity when compared to the non-robust version, GMM-DAE.
> It can also be seen that enforcing coupling between the latent representations of the original and adversarial samples (Ours) leads to better performance than simply augmenting them (Augmented).  It is worth pointing out that the GMM-DAE maintains better performance than a standard VAE model (please refer to Table 1 in the main paper, MNIST VAE results).
>     This is due to the well-structured latent space in GMM-DAE. This observation further confirms our hypothesis in Section 3 that when similar-looking samples are modelled together in the latent space of the model, robustness can be improved.
>
>
> 2.  Robustness to the downstream task - We would like to point out that the network and implementation details of the classifier network used in this section are provided in Section A.4 of the Appendix (lines 101-103).
>
>  3.  Computational complexity - The proposed method is comparatively less expensive than the robust VAE baselines considered.
> As we have pointed out in the limitations section of the main paper, the method is still expensive when compared to the non-robust model by Saseendran et.al.
> We will further clarify this point in the abstract of the paper.
>
> 4. Additional evaluation of decoder image quality - As suggested, we have evaluated the MSSSIM between the reference images $x_r$ and its reconstructions ${x_r} $ tilde and the adversarial images $x_a$ and its corresponding reconstructions ${x_a}$  tilde for both attack modes.
> The results are reported in Table 4 in Section A.7 in the Appendix.
> Compared to the non-robust variants (VAE, $\beta$-VAE, $\beta$-TCVAE), the quality of the reconstructions of the reference images is compromised in robust VAE models (LipschitzVAE, SE, AVAE), whereas the proposed model exhibits comparatively better performance.
> This further aligns with the observation that our model yields better reconstruction fidelity than all the baselines. (Please refer to the comment section of Reviewer qDc3 (R1) for more details.)
> Further, we observe the higher similarity between the adversarial images and their reconstructions for all robust VAE models.
> This is due to the fact that all these models employ adversarial training.
> Thank you for suggesting this new line of evaluations of our model.
> We will add these results to the Appendix of the camera-ready version.
>
> 5. Exploring unexplored regions via adversarial samples -
> In our understanding, from the perspective of the attacker, adversarial examples are optimized to be close to the decision boundaries and thus tend to end up in unexplored regions of the latent space in *non-robust* networks.
> Meaning that adversarial samples are encoded to latent representations that are not in close proximity to the learned latent representations of the original training data.
> However, from the perspective of building robust models, even adversarially optimized samples should follow the same distribution as benign samples so that the model can make a robust, correct decision. There might still be unexplored regions of the latent space, in which, for example samples from new, previously unseen classes might end up. Yet, epsilon perturbations of the data are regularized to not end up in such regions. We will clarify this aspect in the paper.

---

### Official Review · Reviewer_Pa7M · 2022-07-12

**Rating:** 6
**Confidence:** 4
**Soundness:** 3 good
**Presentation:** 3 good
**Contribution:** 3 good

**Summary:**

This paper proposes a technique to improve the robustness of VAEs. It builds upon earlier work by Saseendran et al, Ghosh et al. which provide a deterministic alternative to the variational formulation of VAEs. The formulation by Saseendran et al. uses the Kolmogorov-Smirnov (KS) test to regularize the latent space of deterministic auto encoders and impose a multi modal gaussian prior.
This paper extends this idea by imposing the same multi modal gaussian prior on the pair ($z_n$, $z^{adv}_n$) where $z^{adv}_n$ is the adversarial latent code obtained via perturbation at the input image level (added before the encoder). This is as opposed to just imposing the prior on $z_n$ like in earlier work.
This means that the regularizer goes from: the one point KS test used in Saseendran et al. to a two sample KS test between latent codes and their adversarial versions.
The models used in the paper are VAE, beta VAE, Smooth Encoder, AVAE etc evaluated on MNIST, FashionMNIST, SVHN, CelebA.
There are two attack vectors considered:
1. Latent space attack: adding perturbation to an image such that its latent code moves close to the latent code of a target image.
2. Maximum damage attack: adding perturbation to an image such that its reconstruction its now far away from its vanilla reconstruction
Results show that there is consistent improvement in robustness across the board. This is evidenced by close similarity scores between image reconstructions.

**Questions:**

1. A it stands, the paper does not have a single image about what samples look like or even what adversarial example would look like for the model. I think having pictures for these kind of papers make them read better.
2. I found the paper fairly clear overall except the part in the evaluation subsection of Section 4, where the authors introduce a bunch of symbols at the same time. I think having clear definitions for each of reference image ($x_r$), target image ($x_t$) and adversarial image ($x_a$) in addition to clarifying the relationship between each of them would clear things up. (Maybe a diagram/image?)
3. It seems that the number of modes in dataset has to match those in the prior. I would like to see some comments on this from the authors. What happens if the number of modes don’t match? We know from Saseendran et al. that image quality drops but does robustness also fall in a similar fashion? Or maybe robustness falls faster or slower than image quality?

Typo:
Ref [26] does not have paper name. Listed several times in the paper. Key reference.


**Limitations:**

I think the authors should list that the number of modes in the prior has to match the number of modes in the dataset as a limitation.

**Strengths And Weaknesses:**

Strengths:
1. Comprehensive Evaluation - The paper uses a variety of datasets to prove their point. A minor criticism could be that larger datasets are not considered (which is valid), however, datasets used in the paper are also used by prior work.
2. Problem is well motivated and the exposition is clear
3. Improving robustness in generative models is an important problem so this result is significant. Experiments show large improvements in FID so the results seem solid.


Weaknesses:
1. Unclear scope - How to pick modes to match the prior? What happens if they don't match?
2. Section 4 seems a little unclear in an otherwise well written paper.
3. The paper builds upon prior work in the area so the novelty of the technique is not very high. Having said that, it is an application of prior work into a new area which makes this point a less important.

---

> ### Author Response · Authors · 2022-08-02
> **Sensitivity analysis of the number of modes and robustness**
>
>  1. Sensitivity analysis of the number of modes in the chosen GMM prior - As pointed out correctly the number of modes in the
> chosen prior is a hyperparameter of our model. We agree that a
> sensitivity analysis of the number of modes of the GMM prior and the
> observed robustness of the model is an interesting experimental
> analysis. We report the robustness and generation performance of our
> model on MNIST images for different number of components in the chosen
> prior in Table 2, Section A.5 in the Appendix. We use the same number of modes
> used in Saseendran et.al for all our experiments. As expected
> from previous analysis in Saseendran et.al., with an increased number of
> modes in the GMM prior, the generation performance of our extended model is
> also improved. Most importantly, we observe a similar trend for
> robustness as well. That is, with a higher number of components in the
> chosen GMM prior the model exhibits improved robustness. Thank you for
> suggesting this line of experiments. We will add the results of the
> sensitivity analysis to the paper and discuss this point in the
> limitations section.
>
> 2. Improving clarity in the evaluation subsection - In the
> experimental section of the paper the variables $x_r$ and
>  $x_a$ denotes the reference image and the perturbed variant
> of the input image for two types of attacks in VAEs. The definition of
> the reference image varies with the mode of attack.
>
>       1.  In maximum damage attacks, the reference image $x_r$ is
>     simply an original input image provided to the model.
>
>       2.  For latent space attacks, the reference image $x_r$ is
>     chosen by an adversary to attack the model, also referred to as
>     target image.
>
>    We agree that the notation in this section is rather involved. In the
>     camera-ready version of the paper, we will move the qualitative analysis
>    from Section A.3 in the Appendix to the main paper. With the help of
>     image samples, we will clarify the definitions of reference and
>     adversarial images for both attacks. We will also consider simplifying
>     the notation.
>
> Thank you for noticing the missing title in the reference \[26\]. We
> have added it in the revised version of the paper.

---

> > ### Comment · Reviewer_Pa7M · 2022-08-08
> > **Number of modes**
> >
> > Thank you for the extra experiments.
> >
> > I was expecting results to show a maxima around the number of modes in the dataset (10 for mnist) or at least level off once they reach the actual number of modes in the dataset. Saseedran et al. seem to show similar results so I'm not sure if I'm missing something.
> >
> > Having said that, the robustness following a similar trend to image quality is expected and it is nice to have some extra confirmation.
> >
> > I think this improves the paper slightly and still maintain my score.

---

> > > ### Author Response · Authors · 2022-08-09
> > > **Further remarks**
> > >
> > > The general trend observed in Saseendran et.al is that the FID improves when the number of modes in the prior is increased. It can be also observed from their sensitivity experiments, that even when the number of modes is further increased from 10, the generation performance is further improved.
> > >
> > > In the present paper, we use a fully connected architecture for training on MNIST and FASHIONMNIST images in contrast to the convolution architecture in Saseendran et.al. (Please refer to Section A.4 in the Appendix for architectural details.)
> > > In our experiments, we observe large improvements in terms of robustness when the number of modes is increased from 1 to 10, and a slight improvement when the number of modes is increased further. Although we do not observe a large improvement in terms of the generation when increasing the number of components from 1 to 10, the FID score still drops from $42.45$ to $39.37$. We assume that this effect is less significant in our work compared to Saseendran et.al. due to the architectural choices.

---

> > > > ### Comment · Reviewer_Pa7M · 2022-08-09
> > > > **Response**
> > > >
> > > > Got it, thank you for the clarification.

---

### Official Review · Reviewer_qDc3 · 2022-07-19

**Rating:** 6
**Confidence:** 2
**Soundness:** 2 fair
**Presentation:** 3 good
**Contribution:** 3 good

**Summary:**

This paper studies the robustness of deterministic autoencoders by introducing a regularization scheme to incorporate adversarially perturbed data points into the training pipeline. The proposed method increases the generation and robustness of the learned representations without increasing the computational complexity.

**Questions:**

About the sentence in lines 45 and 46, ``However, regularizing the VAE objective ..., Hence, ... ''. The traditional methods lead to poor reconstruction fidelity, while you aim at maintaining the generation performance? Not both the reconstruction and generation quality of the image?

**Ethics Review Area:**

["I don’t know"]

**Limitations:**

Yes

**Strengths And Weaknesses:**

Strengths:

1. The writing is good, and the paper is easy to follow.
2. The experiment part demonstrates the proposed method has a large improvement in robustness of the most time.

Weaknesses:

In the paper, the author may need to show some qualitative results.

---

> ### Author Response · Authors · 2022-08-02
> **Additional evaluation on the quality of the reconstructed images**
>
>  1. Maintaining both reconstruction and generation fidelity - The robust VAE models sacrifice generation performance for improved robustness. That is, the sampling FID is compromised in these models. Hence, in our model, we seek to focus on improving the robustness of autoencoders while still maintaining the generation performance. We have clarified this point in the revised version of the paper. To showcase that our model effectively maintains both reconstruction and sampling FID, we have benchmarked its reconstruction fidelity against important baseline models on MNIST and FASHIONMIST images, please refer to the Table 1 below. Our results show that the reconstruction performance is also compromised in the robust VAE models when compared to the non-robust variants. However, the proposed model maintains both reconstruction and sampling FID when compared to the baseline models.
>  Method       | MNIST(↓)  | FASHIONMNIST(↓) |
> |:-------------|:---------:|:---------------:|
> | VAE          |   25.10   |      41.23      |
> | *β*-VAE      |   26.21   |      41.98      |
> | *β*-TCVAE    |   26.19   |      41.23      |
> | LipschitzVAE |   30.15   |      47.21      |
> | SE           |   28.80   |      43.31      |
> | AVAE         |   27.29   |      42.98      |
> | Ours         | **19.91** |    **35.74**    |
>
> Table 1: Reconstruction fidelity of MNIST and FASHIONMNIST images.
>
> 2. Ethical review flag  -  Our paper proposes a method to improve robustness in autoencoders.
> We could not see any immediate ethical concerns arising from this fundamental research.
> Upon the suggestion of the other reviewers, we have included a discussion on the potential societal impact of our paper in the common comment section above.
> Could you please specify your concern regarding the ethics flag?

---

### Author Response · Authors · 2022-08-02
**Response to common questions from reviewers**

We would like to thank the reviewers for the generally positive reviews
of our paper and for their constructive remarks. We will begin with
answering common questions that were raised by multiple reviewers and
afterwards answer individual questions. We uploaded a revised version of
the main paper with the following corrections:

1.  We have edited Figure 1 and its title to enhance the readability.

2.  We have fixed all the typos and clarifications suggested by the
    reviewers.

We also uploaded a revised Appendix pdf in the supplementary section with the following quantitative results:

1.  A sensitivity analysis of the number of modes in the GMM prior.

2. A sensitivity analysis of the coupling strength parameter.

3. Additional evaluation of decoder quality.

**Common questions from the reviewers**

1.  Potential societal impact - Variational autoencoders enable
learning meaningful representations of complex high dimensional data
without any supervision. This further enhances the usability of these
learned representations for various downstream tasks and applications
where limited data is available for example, due to privacy/security
concerns. Hence, it is important to study the robustness of these
representations along with their accuracy, especially when employed in
real-world applications. In the present paper, we propose a method to
train robust VAEs with high fidelity in the learned latent space. Since
our work is a step towards more robust models, we hope to see a positive
social impact. There is currently limited work in this direction and we
believe that our method encourages potential future work in developing
robust VAE models. On the other hand, we should take into consideration
the possible negative social impact of this research, especially in
safety-critical applications. Although we observe superior robustness in
our model against the existing attacks, similar performance cannot be
guaranteed on newly discovered attacks on VAEs. Hence, when deployed in
real-world applications we highly recommend testing the model
continuously against newly designed attacks. We also urge the machine
learning community to pursue this work responsibly to enable potential
future research without any misuse. We kindly ask the reviewers if there
are any specific topics to be discussed in this section that we have
missed.

2. Qualitative analysis - We would like to point out that  we provide visual results for MNIST, FASHIONMNIST, SVHN, and
CELEBA images for both adversarial attacks in Section A.3. of the Appendix.
Due to the lack of space, we have not yet added a qualitative analysis to the initial version of the main paper.
Upon acceptance, we will move the CELEBA image samples to the additional page that is allowed for the camera-ready version of the paper.

---

### Author Response · Authors · 2022-08-07
**Discussion period nearing a close**

With the rebuttal discussion period approaching a close, we kindly ask the reviewers whether our answers and additional experiments have clarified their questions. Please let us know if any further queries or concerns.

---

### Meta-Review · Area_Chair_vtpJ · 2022-08-30

**Recommendation:** Accept
**Confidence:** Certain

**Metareview:**

This paper received generally positive reviews that, after discussion, all backed acceptance.  The paper was praised for its empirical evaluations, potential significance, clarity, and applicability.  While some questions and lower-level issues were raised, I do not feel that the reviewers raised any significant issues that would be a barrier to acceptance, with the small number of issues that were raised well addressed by the authors' responses.

My own personal view of the work is also very positive (perhaps more so than the reviewers themselves): I think this is strong work that will be of significant interest to the community.  The empirical results are a particular highlight, both in terms of the performance shown, the comprehensive set of experiments considered, and the numerous and appropriate baselines compared to.  While I do have some suggestions and minor gripes (see below) that I would like addressed in the final version of the paper, I have no hesitation in enthusiastically recommending its acceptance.

Suggestions and minor issues:
- My most important complaint with the paper is that the title is too strong and overclaiming.  It suggests a trade-off will never occur and that the result applies to all deterministic auto-encoders, rather than the specific type considered.  I do not think either of these are true: just because robustness has been improved with minimal change in FID score, does not mean there will not be a trade-off with future developments (i.e. there may well be ways of improving the image quality of [26] that would no longer give good robustness when combined with the suggested approach). Please, therefore, change the title to something more measured and precise for the camera-ready paper.
- It would be good to provide more explicit timing information about the training times of all the different models, rather than just SE.  Claims are made in the intro and conclusions about speed, but, unless I have missed something, I did not really feel these were properly supported.
- While I think the paper is mostly quite clear and well written, I do think the writing could be improved in some of the key technical sections; I generally found Section 3 to be the worst written part of the paper.  In particular, I think more high-level explanation was required.  While the maths itself is not too difficult to follow, it took me quite a few reads through to get a feel for the intuitions.  To give an example of a specific issue, the right-hand side of Figure 1 comes too early and lacks context: the reader will naturally assume that they should be able to understand what is happening when the figure is first referenced, but actually they need to get to Section 3.3 first to get an idea of what is going on.
- It might be useful for the authors to have a look at https://openreview.net/forum?id=nzvbBD_3J-g because it actively argues against using GMM priors in the more conventional VAE setting, on the basis that such inductive bias can be more effectively be incorporated through a customised decoder architecture (and not treating the latents as the representation itself), than through regularisation.  Of course, their insights may well not carry over to the deterministic auto-encoder setting, but it does hint at an interesting alternative approach and may be worth discussing or at least acknowledging.
- Please increase the text size in the figures, these are very difficult to see at present.

**Award:**

No

---

### Decision · Program_Chairs · 2022-09-14

Accept